# Validating modeled critical crack length for crack propagation in the snow cover model SNOWPACK

Bettina Richter[1], Jürg Schweizer[1], Mathias W. Rotach[2], and Alec van Herwijnen[1]

[1]WSL Institute for Snow and Avalanche Research SLF, Davos, Switzerland
[2]Institute for Atmospheric and Cryospheric Sciences, University of Innsbruck, Innsbruck, Austria

**Correspondence:** Bettina Richter (bettina.richter@slf.ch)

**Abstract.** Observed snow stratigraphy and snow stability are of key importance for avalanche forecasting. Such observations are rare and snow cover models can improve the spatial and temporal resolution. To evaluate snow stability, failure initiation and crack propagation have to be considered. Recently, a new stability criterion relating to crack propagation, namely the critical crack length, was implemented into the snow cover model SNOWPACK. The critical crack length can also be measured in the field with a propagation saw test, which allows for an unambiguous comparison. To validate and improve the parameterization for the critical crack length, we used data from three years of field experiments performed close to two automatic weather stations above Davos, Switzerland. We monitored seven distinct weak layers and performed in total 157 propagation saw tests on a weekly basis. Comparing modeled to measured critical crack length showed some discrepancies stemming from model assumption. Hence, we replaced two variables of the original parameterization, namely the weak layer shear modulus and thickness, with a fit factor depending on weak layer density and grain size. With these adjustments, the normalized root mean square error between modeled and observed critical crack lengths decreased from 1.80 to 0.28. As the improved parameterization accounts for grain size values of critical crack lengths for snow layers consisting of small grains, which in general are not weak layers, become larger. In turn, critical weak layers appear more prominently in the vertical profile of critical crack length simulated with SNOWPACK. Hence, minimal values in modeled critical crack length better match observed weak layers. The improved parameterization of critical crack length may be useful for both weak layer detection in simulated snow stratigraphy as well as providing more realistic snow stability information - and hence may improve avalanche forecasting.

## 1 Introduction

Snow slab avalanches are hazardous and can threaten people and infrastructure. Each year, around a 100 avalanche fatalities occur in the European Alps (Techel et al., 2016). Whether avalanche release is likely, largely depends on snow layering, in particular the complex interaction between slab layers and a so-called weak layer (Schweizer et al., 2008). Such weak layers often form near or at the snow surface and, if subsequently covered by a snowfall, can sometimes persist throughout the season.

Dry-snow slab avalanches start with a failure in the weak layer resulting in a macroscopic crack. If this crack reaches a critical size, the crack will rapidly propagate outward (e.g. McClung and Schweizer, 1999; Schweizer et al., 2003a; van Herwijnen and Jamieson, 2007), provided the tensile strength of the slab allows for crack propagation (Reuter and Schweizer, 2018).

After crack propagation, the slab comes into frictional contact with the bed surface (Simenhois et al., 2012; van Herwijnen and Heierli, 2009), and slope angle mainly determines if an avalanche releases. Snow cover stratigraphy is thus considered an important contributing factor in avalanche forecasting (Schweizer et al., 2003a). To assess snow instability therefore requires information on the spatial distribution of slab and weak layer properties and how easily cracks form and propagate.

Snow stratigraphy information is traditionally obtained with manually observed snow profiles, where each layer is characterized by grain type, grain size and hand hardness (Fierz et al., 2009). Manually observed snow profiles are often completed with snow stability tests (e.g. Schweizer and Jamieson, 2010). However, information on snow stratigraphy and snow stability are rare point observations which are very time consuming and sometimes dangerous to obtain. Numerical snow cover models can help increase the spatial and temporal resolution of information on snow stability (e.g. Lafaysse et al., 2013).

Crocus (Brun et al., 1992; Vionnet et al., 2012) and SNOWPACK (Lehning et al., 2002; Wever et al., 2015) are detailed snow cover models which also provide stability indices (Schweizer et al., 2006; Lehning et al., 2004; Vernay et al., 2015). The French model chain SAFRAN–SURFEX/ISBA-Crocus–MEPRA (S2M) predicts indices describing the avalanche danger at regional scale (Durand et al., 1999; Lafaysse et al., 2013). Crocus is driven with input of the meteorological model SAFRAN and the stratigraphy on virtual slopes for a range of elevations and aspects are simulated. The expert system MEPRA combines various stability indices with a set of rules to evaluate the simulated snow stratigraphy in terms of stability classes and derives the avalanche danger (Giraud and Navarre, 1995). However, model predictions such as the avalanche danger level are difficult to validate (Schweizer et al., 2003b).

The snow cover model SNOWPACK is driven with data from automatic weather stations. Stability indices are then calculated from modeled snow stratigraphy, i.e. modeled layer properties. Several stability indices have been implemented in SNOWPACK, in particular the natural stability index SN38 and the skier-stability index SK38 (Lehning et al., 2004; Monti et al., 2016). Both stability indices relate to failure initiation and are based on the ratio of the shear strength of a weak layer to the load of the overlaying slab and, for SK38, the approximate stress due to a skier (Föhn, 1987; Jamieson and Johnston, 1998). Weak layer shear strength is parameterized from shear frame measurements in relation to snow density and grain type (Chalmers, 2001; Jamieson and Johnston, 2001). Shear strength and related stability indices are calculated in SNOWPACK for each modeled snow layer (Lehning et al., 2004). To validate these stability indices, previous studies relied on a variety of field measurements, including shear frame measurements, stability tests, manual snow profiles and avalanche observations, to compare modeled stability metrics with observations. Whereas SK38 is closely related to avalanche activity, SN38 is a rather poor predictor of natural avalanche release (Gauthier et al., 2010). While modeled SK38 performed poorly in terms of identifying potential weak layers, combining it with structural parameters, e.g. differences in grain sizes or hand hardness, the performance improved (Schweizer et al., 2006; Schweizer and Jamieson, 2007).

Recently, a parameterization for the critical crack length, which relates to the onset of crack propagation, was suggested by Gaume et al. (2017) and implemented into SNOWPACK. The critical crack length can directly be measured in the field with the propagation saw test (PST; Gauthier and Jamieson, 2008a), which greatly facilitates the validation. A qualitative comparison suggested that local minima in modeled critical crack lengths for one particular field day agreed with observed critical crack lengths (Gaume et al., 2017). Schweizer et al. (2016) monitored the temporal evolution of a weak layer during the winter

season 2014-2015 above Davos, Switzerland. They compared the temporal evolution of the critical crack length observed with PST experiments to the critical crack length predicted by SNOWPACK. Although SNOWPACK reproduced the overall trend fairly well, the seasonal increase was too pronounced. They attributed these discrepancies to an overestimation of weak layer density in SNOWPACK, however their analysis only included one weak layer of faceted crystals.

In this study, we will investigate the performance and limitations of the SNOWPACK model to predict the critical crack length. We will use a dataset containing weekly field measurements. During three winter seasons, 2014-2017, we tracked persistent weak layers with time at different locations close to an automatic weather station and conducted measurements of critical crack length. This dataset was used to validate and improve the parameterization of the critical crack length suggested by Gaume et al. (2017). The new formulation allows for a better representation of the temporal evolution of the critical crack

length. It reduces the dependency on weak layer density and considers the microstructural parameter grain size. Minima in the vertical profile of the critical crack length corresponded to associated weak layers, which may improve weak layer detection.

## 2 Methods

### 2.1 Field sites

We collected data during three winter seasons, from 2014-2015 to 2016-2017, at two flat field sites above Davos, Switzerland.

Both sites are relatively sheltered from wind and equipped with an automatic weather station (AWS) measuring snow depth, air temperature, relative humidity, wind speed, wind direction, incoming and outgoing short- and longwave radiation. The Weissfluhjoch site (WFJ; 46.830° N, 9.809° E) is located at 2536 m a.s.l. and the Wannengrat site (WAN7; 46.808° N, 9.788° E) is located at 2442 m a.s.l. about 3 km to the southwest from WFJ; they typically have a similar snowpack.

### 2.2 Snow profiles and stability tests

At both sites, manual snow profiles were recorded on an almost weekly basis between January and March (Table 1). Data on hand hardness, temperature, density, grain type and grain size were recorded according to Fierz et al. (2009). Density was measured either for each individual snow layer with a density tube (volume of 100 cm$^3$, 3.7 cm inner diameter) or every 3 cm in a vertical profile using a density cutter (box-type density cutter of 100 cm$^3$, 6 cm × 3 cm × 5.5 cm Proksch et al., 2016). The density of layers thinner than 3 cm could therefore not be measured.

Manual snow profiles were complemented with stability tests, namely the Compression Test (CT; van Herwijnen and Jamieson, 2007), the Extended Column Test (ECT; Simenhois and Birkeland, 2009) and the Propagation Saw Test (PST; Gauthier and Jamieson, 2006; Sigrist and Schweizer, 2007; van Herwijnen and Jamieson, 2005). CTs and ECTs were conducted to identify weak layers. The PST is a fracture mechanical field test and was used to assess the critical crack length required for rapid crack propagation in an a priori known weak layer. It consists of an isolated column of 30 cm width and

a variable length of at least 120 cm (Figure 1). A failure in the weak layer is initiated by cutting the weak layer with a snow saw until the crack propagates. The length at which the crack propagated is called the critical crack length $r_c$. The critical

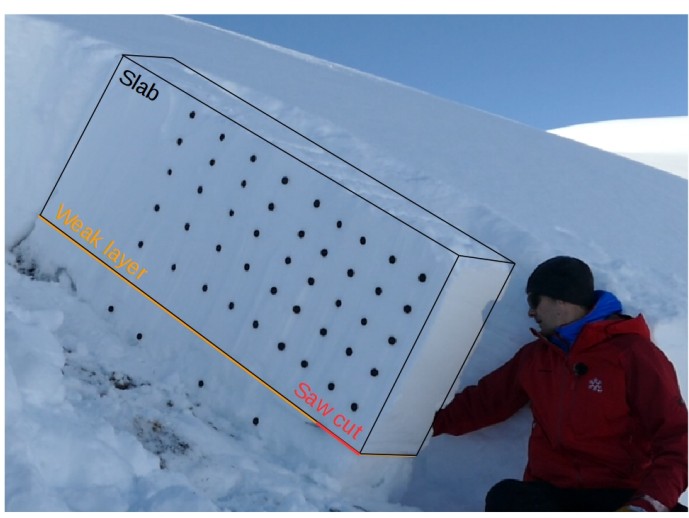

**Figure 1.** Picture of a Propagation Saw Test (PST). Schematic representation of the slab (black lines) and the weak layer (orange line). Red line indicates the artificial crack, initiated with a saw. Photo credit Julia Wessels.

**Table 1.** Overview of number of weak layers and Propagation Saw Test (PST) results available for validation; data collected during three winter seasons and at two field sites: Weissflujoch (WFJ) and Wannengrat (WAN7).

| Year | Field site | Number of persistent weak layers | Number of field days (Jan-Apr) | Number of PST experiments |
|------|-----------|----------------------------------|--------------------------------|---------------------------|
| 2014-2015 | WAN7 | 2 | 8 | 44 |
| 2015-2016 | WAN7 | 1 | 9 | 33 |
| 2015-2016 | WFJ | 2 | 8 | 28 |
| 2016-2017 | WAN7 | 1 | 10 | 33 |
| 2016-2017 | WFJ | 1 | 14 | 19 |

crack length as well as the propagation distance are recorded. On each of the 49 measurement days (Table 1), we performed CTs, ECTs, and one to five PSTs per weak layer. In total, 157 PST experiments were conducted in 7 different weak layers. We calculated average $r_c$ values from PSTs conducted in any given weak layer on a particular day. This yielded a dataset of 61 averaged critical crack lengths, which was then compared to $r_c$ values simulated with SNOWPACK for the associated layers.

5 Weak layers were coded after their grain type (GT) according to Fierz et al. (2009) and burial date (YYMMDD) with a code GTYYMMDD.

## 2.3 SNOWPACK

We used the snow cover model SNOWPACK (version 3.5.0, revision 1801) to simulate the snow stratigraphy (Bartelt and Lehning, 2002). The model was driven with AWS data at both sites, using air temperature, relative humidity, snow surface temperature, wind speed, short- and longwave radiation. For the WAN7 site the snow cover mass balance was enforced with the increment of measured snow depth. For the WFJ site additional data from a heated rain gauge was used to estimate the occurrence of rain (WSL Institute for Snow and Avalanche Research SLF, 2015). At the field site WAN7, the sensor measuring the snow surface temperature malfunctioned. We therefore chose to use Neumann boundary conditions at the snow surface at both sites to estimate the snow surface temperature from energy fluxes (Bartelt and Lehning, 2002; Lehning et al., 2002). At the bottom of the snowpack, a constant geothermal heat flux of $0.06\,\mathrm{W\,m^{-2}}$ was assumed (Davies and Davies, 2010; Pollack et al., 1993). The simulation time step was 15 min and the output was stored daily at 11 UTC, which corresponds approximately with the times of manually observed snow profiles (i.e. between 9 and 14 UTC). Hence, the comparison to measurements was not affected by daily variations, which are generally low.

Based on these meteorological input data, SNOWPACK simulates the formation and metamorphism of snow layers. Each layer therefore has different properties, mainly characterized by its density, temperature, grain type and grain size. To compare observed weak layers with the corresponding simulated weak layers, we stored the deposition date of simulated layers. Each modeled snow layer was assigned within the SNOWPACK model with a deposition date corresponding to the date when a new layer was defined in the model. For observed weak layers, however, we only know the burial date, i.e. the day when a weak snow surface was covered by new snow. To match observed weak layers with the corresponding simulated layer, we therefore searched for the simulated layer, which was deposited immediately before the burial date of the observed weak layer. In other words, we identified the simulated weak layer by choosing the uppermost simulated layer with a deposition date older than the burial date of the observed weak layer. Layers of surface hoar are treated separately in SNOWPACK. Since surface hoar forms by deposition of water vapor from the air on the snow surface, and not from precipitation, it is only treated as snow layer within SNOWPACK, if certain conditions are fulfilled during burial (Lehning et al., 2002). Thus, modeled surface hoar only "becomes" a snow layer at burial. For observed layers of surface hoar, we therefore first checked whether the layer which was covered by new snow also consisted of surface hoar in the simulation. To temporally track this layer of modeled surface hoar, we then identified the simulated weak layer by choosing the lowermost simulated layer with a deposition date equal to the burial date of the observed layer. All layers above an associated weak layer were assigned to the slab. To obtain slab thickness, layer thicknesses of all simulated slab layers were summed up. Slab density was obtained by a thickness-weighted average of simulated slab layers.

From simulated layer properties, snow mechanical properties required for the parameterization of the critical crack length (see Section 2.4) are computed in SNOWPACK. As suggested in Gaume et al. (2017), the elastic modulus of the slab, $E$, was related to the slab density $\rho_{sl}$ by a power law fit of the data collected by Scapozza (2004):

$$E = 5.07 \times 10^9 \left( \frac{\rho_{sl}}{\rho_{ice}} \right)^{5.13} \mathrm{Pa,} \tag{1}$$

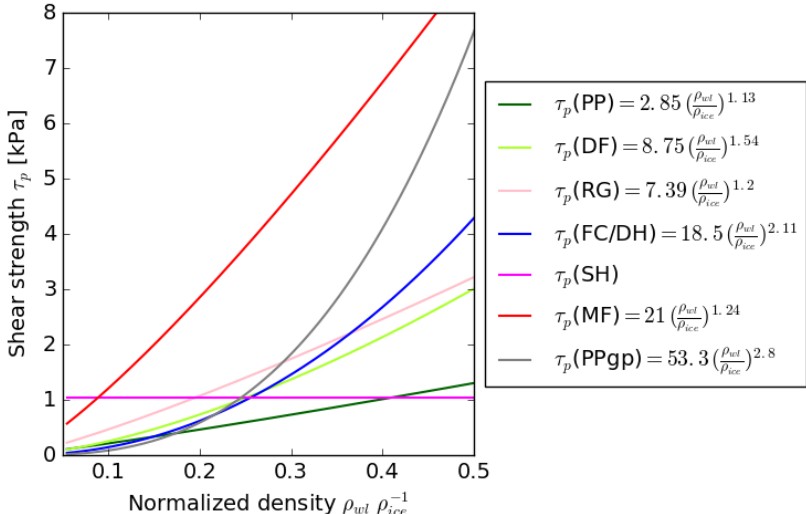

**Figure 2.** Parameterization of the shear strength $\tau_p$ as implemented in SNOWPACK. Except for surface hoar (SH), $\tau_p$ is a power law function of normalized density $\tau_p = a \left( \frac{\rho_{wl}}{\rho_{ice}} \right)^b$. Values for $a$ and $b$ depend on grain type. Grain types are precipitation particles (PP), decomposed and fragmented precipitation particles (DF), rounded grains (RG), faceted crystals (FC) and depth hoar (DH), surface hoar (SH), melt forms (MF) and graupel (PPgp).

with $\rho_{ice} = 917\,\mathrm{kg\,m^{-3}}$ the density of ice. For the original parameterization of $E$, a correlation coefficient of 0.9 was reported (Scapozza, 2004). We used the default implementation for the shear strength $\tau_p$ in SNOWPACK, which depends on grain type (Figure 2). For all grain types except for SH (see caption of Fig. 2 for the acronyms of different grain types), $\tau_p$ solely depends on weak layer density $\rho_{wl}$ through a power law function $\tau_p = a \left( \frac{\rho_{wl}}{\rho_{ice}} \right)^b$. Values for $a$ and $b$ were derived for different grain

5    types based on shear frame measurements; correlation coefficients of 0.31 to 0.54, depending on grain type, were reported (see Table 8 in Jamieson and Johnston, 2001). For SH, the parametrization of Lehning et al. (2004) was applied, which is a function of age of the weak layer, the normal stress $\sigma_n$, slab thickness $D_{sl}$, snow depth (HS), weak layer thickness $D_{wl}$ and weak layer temperature $T_{wl}$. The normal stress $\sigma_n = \rho_{sl} g D_{sl}$ is exerted on the weak layer due to the overlying slab, with the slab thickness $D_{sl}$ and the gravitational acceleration $g$. In Fig. 2 $\tau_p$ is shown for a layer of SH with an age of 7 days, $D_{wl} = 0.01\,\mathrm{m}$,

10    $D_{sl} = 0.5\,\mathrm{m}$, $HS = 1\,\mathrm{m}$, $T_{wl} = -5°\mathrm{C}$, $\rho_{sl} = 200\,\mathrm{kg\,m^{-3}}$ and $\sigma_n = 0.981\,\mathrm{kPa}$.

## 2.4 Critical crack length parameterization

To estimate the critical crack length $r_c$ from snow mechanical properties, we used the parameterization suggested by Gaume et al. (2017). They modeled crack propagation with the discrete element method, using an idealized structure of the weak layer by assembling spheres in a triangular shape. For a flat field site (slope angle $\theta = 0$) $r_c$ reduces to:

$$r_c = \Lambda \sqrt{\frac{2\tau_p}{\sigma_n}}, \tag{2}$$

where the characteristic length scale $\Lambda = \sqrt{\frac{E'D_{sl}D_{wl}}{G_{wl}}}$ includes the plain strain elastic modulus of the slab $E' = \frac{E}{(1-\nu^2)}$, the Poisson's ratio of the slab $\nu = 0.2$, and the shear modulus of the weak layer $G_{wl} = 0.2$ MPa, as suggested by Gaume et al. (2017).

All layer properties required in Eq. (2) are calculated within SNOWPACK. Furthermore, Eq. (2) was also evaluated using profile data as most properties - i.e. $D_{sl}$, $D_{wl}$, $\rho_{sl}$ and $\rho_{wl}$ - were measured directly in the field. Weak layer shear strength and the elastic modulus of the slab, which were not measured, were derived from measured densities using the same parameterizations as those implemented in SNOWPACK (Eq. (1) and Fig. 2).

## 2.5 Model performance measures and weak layer detection

We used different performance measures to validate the parameterization for the critical crack length, as well as layer properties from SNOWPACK, namely density and layer thickness. To measure the linear relationship between a modeled value $y$ and a measured value $x$, we calculated the Pearson correlation coefficient $r_p$. We considered a level of $p < 0.05$ as significant. To quantify errors, we calculated the normalized root mean square error $NRMSE$:

$$NRMSE = \frac{1}{\bar{x}} \sqrt{\frac{\sum_{i=1}^{n} (x_i - y_i)^2}{n}} \tag{3}$$

where $n$ is the number of measurements (e.g. $n = 61$ is the number of mean values for $r_c$ observed with 157 PST experiments per weak layer and day; see Sect. 2.2) and $\bar{x}$ is the mean of the measurements.

To assess whether the parameterization for the critical crack length implemented in SNOWPACK can be used to automatically identify critical weak layers, we investigated whether the five lowest values in the vertical profile of the critical crack length in SNOWPACK corresponded to the critical weak layers tested in the field. This approach consisted of ranking layers in SNOWPACK according to their $r_c$ values in ascending order. First, we checked whether the global minimum in the simulated vertical profile of critical crack length was close to a simulated weak layer that was matched with the observations. If the layer with the lowest critical crack length in SNOWPACK was in the range of $\pm 5$ cm of an associated weak layer, it was counted as a detection ($d$), otherwise as false alarm ($fa$). If we observed multiple weak layers in one profile, we iteratively identified the layers with the next lowest values in the vertical profile of the critical crack length, by excluding a range of $5$ cm each above and below the prior minimum. Detections and false alarms were counted until either all associated weak layers were found or five

minima in the vertical profile of the critical crack length were compared. If associated weak layers were not detected within the five minima, they were considered not detected ($nd$). For each field day $j$, we summed up $d$, $fa$ and $nd$. This procedure allowed us to calculate a detection rate (DR) and a misclassification rate (MR):

$$DR = \frac{\sum_{j=0}^{m} d}{\sum_{j=0}^{m} d + \sum_{j=0}^{m} nd} \tag{4}$$

$$MR = \frac{\sum_{j=0}^{m} fa}{\sum_{j=0}^{m} d + \sum_{j=0}^{m} fa} \tag{5}$$

where $m = 49$ is the number of field days. Note that $d + nd = n$.

## 3 Results

### 3.1 Winter seasons - weather and snowpack

The snow depth was average during 2014-2015 and generally below average for winter seasons 2015-2016 and 2016-2017 (Figure 3). Each winter, one to two pronounced weak layers developed and consistently failed in CT and ECT tests. These persistent weak layers were tracked in PST experiments throughout the season (Table 1). In the following we will give a detailed description of the formation of these weak layers.

**Winter 2014-2015**

The winter started at the end of October with approximately 60 cm of snow. During the calm weather period starting in mid-November, the near-surface snow transformed into a layer of faceted crystals, forming a persistent weak layer that was buried by snow in mid-December (FC141216). A layer of surface hoar, which had formed in the region in mid-January, was buried on 24 January 2015 (SH150124) and was subsequently observed in the traditional snow profile on 28 January 2015 (Figure 3d). During this winter, we only performed measurements at the WAN7 field site. A more detailed description of the weather development and weak layer formation can be found in Schweizer et al. (2016).

**Winter 2015-2016**

In early November, a first snow storm deposited around 30 cm of snow at both field sites. A period of calm weather followed and large temperature gradients transformed the near-surface snow into a layer of depth hoar (DH151201). On 1 December 2015, local observers reported rain up to 2600 m a.s.l., forming a crust on top of DH151201. In mid-December, an additional 20 cm of snow accumulated on the crust, and subsequently transformed into a layer of faceted crystals during a clear weather period. This layer of facets was then covered by snow on 31 December 2015 (FC151231). From January 2016 on, no further prominent weak layer developed. These two persistent weak layers, below and above the crust, were observed at both field sites.

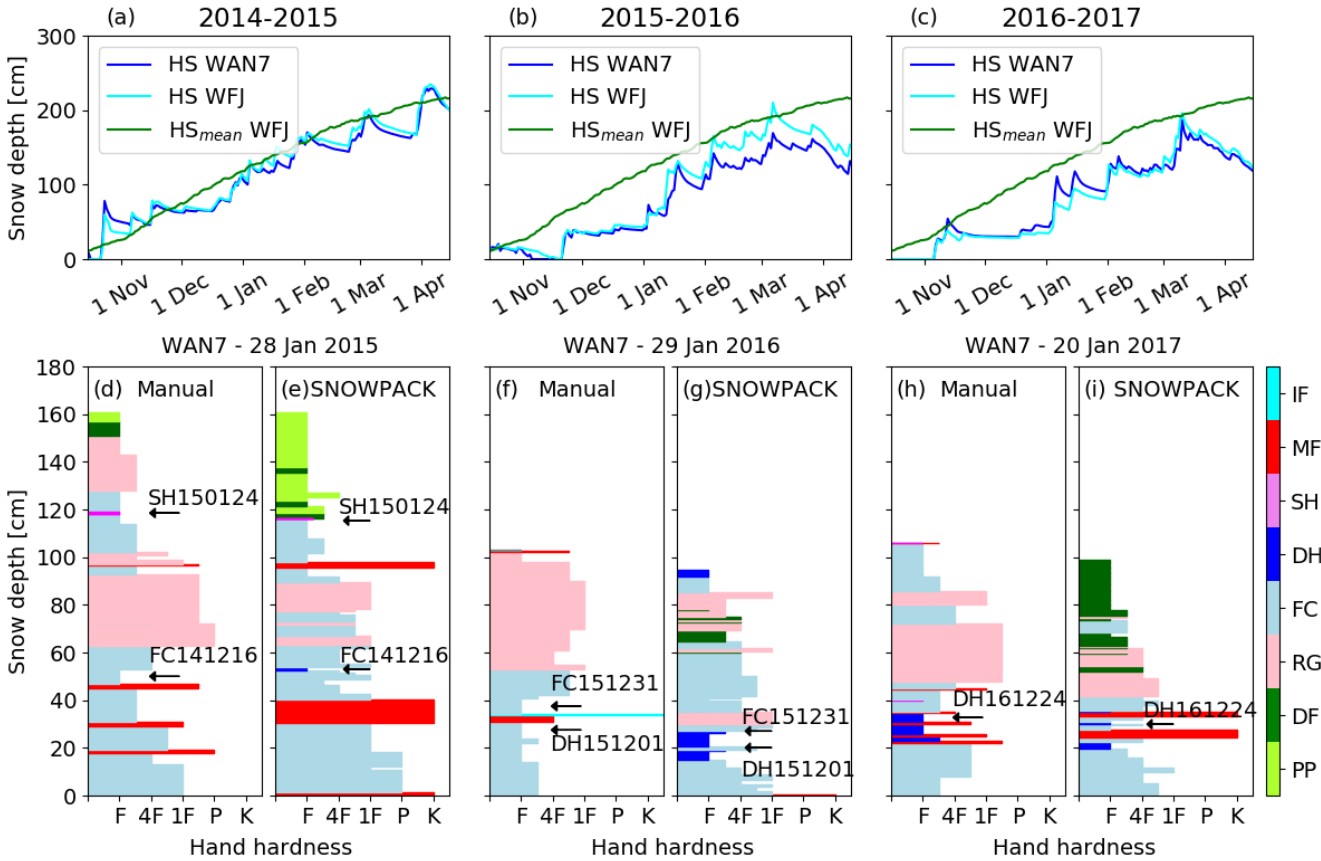

**Figure 3.** Top: Temporal evolution of measured snow depth at both field sites for the winter seasons (a) 2014-2015, (b) 2015-2016 and (c) 2016-2017. $HS_{mean}$ is the measured snow depth at WFJ averaged over 85 years. Bottom: (d,f,h) manually observed snow profile at WAN7 showing hand hardness and grain type (colors) for the end of January each year and (e,g,i) corresponding simulated snow stratigraphy from SNOWPACK. Hand hardness is coded after (Fierz et al., 2009), where F corresponds to fist, 4F to 4 fingers, 1F to one finger, P to pencil, and K to knife. Grain types are precipitation particles (PP), decomposing and fragmented precipitation particles (DF), rounded grains (RG), faceted crystals (FC), depth hoar (DH), surface hoar (SH), melt forms (MF) and ice formations (IF). Arrows with labels indicate critical weak layers which were observed in PST experiments. Labels of weak layers were coded after grain type GT and burial date (GTYYMMDD).

**Winter 2016-17**

This winter was relatively similar to the previous winter, starting with a shallow snowpack followed by a period of calm weather. Around 20-30 cm above the ground, a layer of DH crystals formed. This weak layer was covered by snow on 24 December 2016 (DH161224). Between January and March 2017, several small snow storms occurred such that the snow height reached about 200 cm at the beginning of March. No further pronounced weak layers developed during this winter.

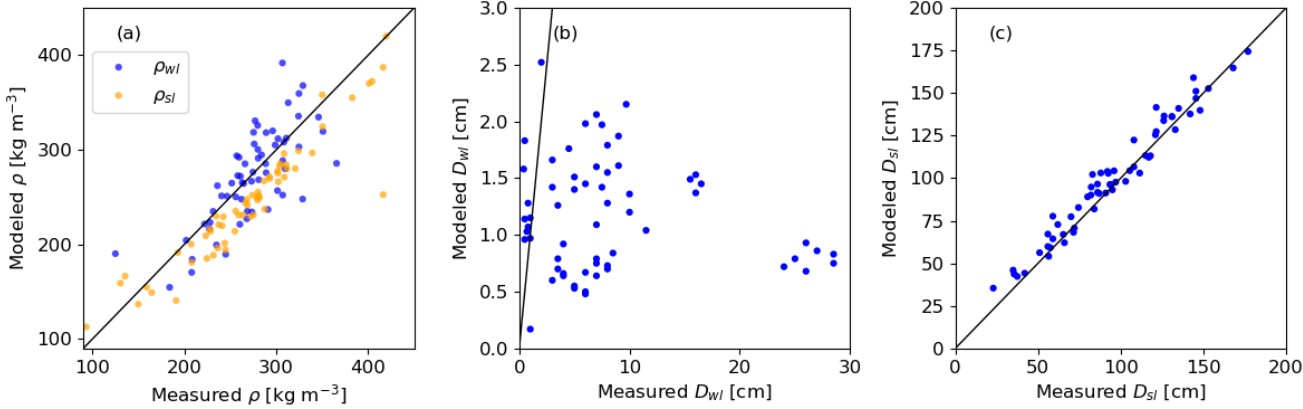

**Figure 4.** Comparison of (a) modeled to measured weak layer density $\rho_{wl}$ and mean slab density $\rho_{sl}$, (b) weak layer thickness $D_{wl}$ and (c) slab thickness $D_{sl}$. Modeled properties were taken from SNOWPACK simulations while measured properties come from manually observed snow profiles. Black line is the 1:1 line.

## 3.2 Modeled snow stratigraphy

For each site and winter season SNOWPACK reproduced the main stratigraphic features reasonably well (Figure 3 d-i). The overall hardness profiles agreed with the observations and the weak layers that were identified and tracked in the field were also present in the simulated profiles. Still, some discrepancies were observed between observation and simulation. One of these
5   discrepancies is the rain crust and ice lense, which formed in the winter 2015-2016 at both field sites (see MF and IF at around 35 cm in Figure 3f), and was used as a reference for the weak layers. SNOWPACK did not simulate this rain crust but rather a thin layer of new snow (Figure 3g), since the 5-m air temperature stayed well below zero degrees.

For the critical crack length parameterization, slab and weak layer properties are required. Most variables in Eq. (2) are related to density, which was also measured in the field. Modeled slab density $\rho_{sl}$ agreed well (Figure 4a) with measured
10   density ($r_p = 0.94$, $p << 0.05$ and $NRMSE = 0.13$), and the agreement for weak layer density $\rho_{wl}$ was only slightly worse ($r_p = 0.75$, $p << 0.05$ and $NRMSE = 0.12$). Modeled slab thickness also agreed well (Figure 4c) with observed $D_{sl}$ ($r_p = 0.98$, $p << 0.05$ and $NRMSE = 0.09$). Weak layer thickness, however, did not agree with observed thickness ($r_p = -0.14$, $p = 0.28$ and $NRMSE = 1.26$). In the field, observers define layer boundaries based on evident differences in layer properties, which is partly subjective, resulting in recorded weak layer thicknesses up to 30 cm. Simulated $D_{wl}$ ranged from 0.17 to 2.52
15   cm, because layer thicknesses were constrained by the simulation time step (Lehning et al., 2002). In contrast, $D_{wl}$ in Eq. (2) described an idealized weak layer thickness closely related to the collapse height after weak layer fracture.

### 3.3 Evolution of the critical crack length

PST experiments were conducted in the persistent weak layers described above. Observed critical crack lengths ranged from 17 to 121 cm and generally increased with time for all sites and seasons (Figure 5). On a single day, repeated PST experiments on the same layer varied by 1 cm to 35 cm resulting in an average relative range of 30 %. Temporary decreases in $r_c$ were sometimes observed after pronounced precipitation events, as for example around 9 March 2017 (Figure 5 d,e). Depending on the weak layer and the field site, seasonal increases in observed $r_c$ were more or less pronounced. For instance, $r_c$ for layer FC151231 only slightly increased from 20 cm to 40 cm at WAN7 (Figure 5), whereas at WFJ the increase was more prominent (Figure 5 b,c). The largest increases in $r_c$ were observed end of March and early April 2017.

The overall temporal trend of $r_c$ (Eq. (2)) was reproduced when using layer properties from SNOWPACK ($r_p = 0.88$, $p <<$ 0.05; Figure 5). However, $r_c$ was generally overestimated ($NRMSE = 1.78$; Figure 6) and simulated $r_c$ values ranged from 4 to 462 cm. The only exception was for a layer of buried surface hoar (SH150124), for which observed and simulated $r_c$ values corresponded well ($r_p = 0.90$, $p = 0.04$ and $NRMSE = 0.41$; Figure 5a). Modeled $r_c$ values (Eq. (2)) were also calculated using layer properties from manually observed snow profiles, if data on thickness and density were available. Doing so, the discrepancies between modeled (Eq. (2)) and observed $r_c$ values were even larger ($r_p = 0.62$, $p << 0.05$ and $NRMSE = 7.11$; Figure 6).

Clearly, the modeled critical crack length with layer properties either from SNOWPACK or from manual snow profiles overestimated observed critical crack lengths, especially later in the season. Since we used the same parameterizations for the required mechanical properties of snow, namely $E$ and $\tau_p$, we investigated differences in modeled and observed density or layer thickness more closely. While modeled slab and weak layer densities as well as slab thickness corresponded well with the observation (Figure 4a,c), modeled and observed weak layer thickness were completely different (Fig. 4b). Indeed, measured values of $D_{wl}$ ranged from 0.4 to 30 cm, whereas in SNOWPACK $D_{wl}$ ranged from 0.2 to 2.5 cm. These differences may be related to difficulties in assessing layer boundaries in manual snow profiles, but are primarily due to numerical boundary conditions limiting the thickness of layers in SNOWPACK. Furthermore, the weak layer shear modulus was taken as constant ($G_{wl} = 0.2$ MPa). This simplification does not account for the temporal evolution of layer properties in the snow cover. Thus, $D_{wl}$ and $G_{wl}$ in the parameterization of Gaume et al. (2017) are likely responsible for the observed discrepancies in modeled critical crack length.

### 3.4 Improvements to $r_c$ parameterization

To improve the $r_c$ parameterization we replaced the ratio $\frac{D_{wl}}{G_{wl}}$ with a parameter $F_{wl} = f(\rho_{wl}, gs_{wl})$, i.e. a function of density $\rho_{wl}$ and the grain size $gs_{wl}$ of the weak layer. $F_{wl} = \frac{D_{wl}}{G_{wl}}$ can be determined from mean $r_{c,obs}$ from PST measurements for each layer and each day in combination with layer properties $\sigma_n$, $E'$ and $\tau_p$ from SNOWPACK using Eq. (2):

$$F_{wl} = r_{c,obs}^2 \cdot \frac{\sigma_n}{2\tau_p} \cdot \frac{1}{E' D_{sl}} \tag{6}$$

Based on the 61 mean observed critical crack lengths in the winters 2014-2015 to 2016-2017 and slab and weak layer variables from SNOWPACK simulations, $F_{wl}$ ranged between $4.03 \times 10^{-9}$ and $1.80 \times 10^{-7}$ m Pa$^{-1}$ (blue dots in Fig. 7). We then fitted

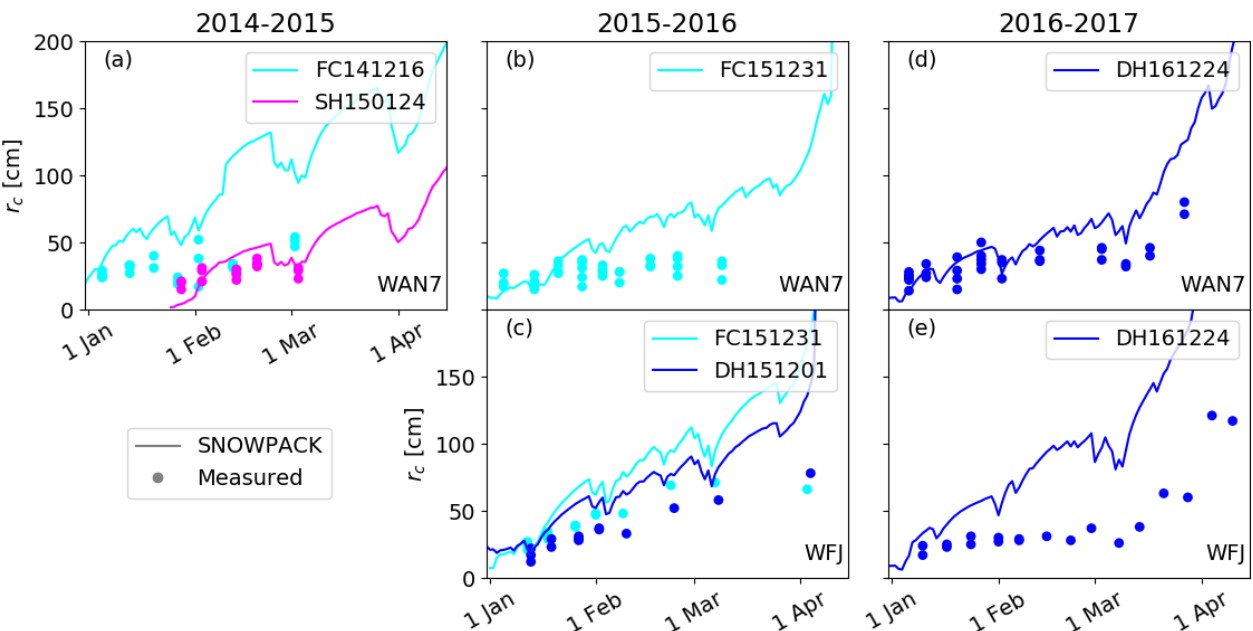

**Figure 5.** Evolution of the critical crack length $r_c$ for the winter seasons 2014-2015 (a: WAN7), 2015-2016 (b: WAN7; c: WFJ) and 2016-2017 (d: WAN7; e: WFJ). Dots represent mean measured $r_c$ values from PST experiments, lines represent modeled $r_c$ values with layer properties from SNOWPACK using Eq. (2).

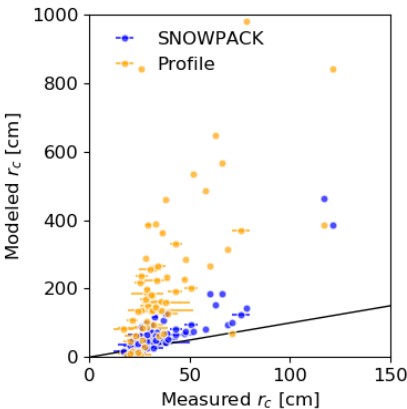

**Figure 6.** Modeled $r_c$ values (Eq. (2)) with layer properties from SNOWPACK (blue dots) and from manual profiles (orange dots) with averaged measured critical crack lengths from PST experiments. Error bars indicate the range of measured critical crack lengths for each point. The black line is the 1:1 line.

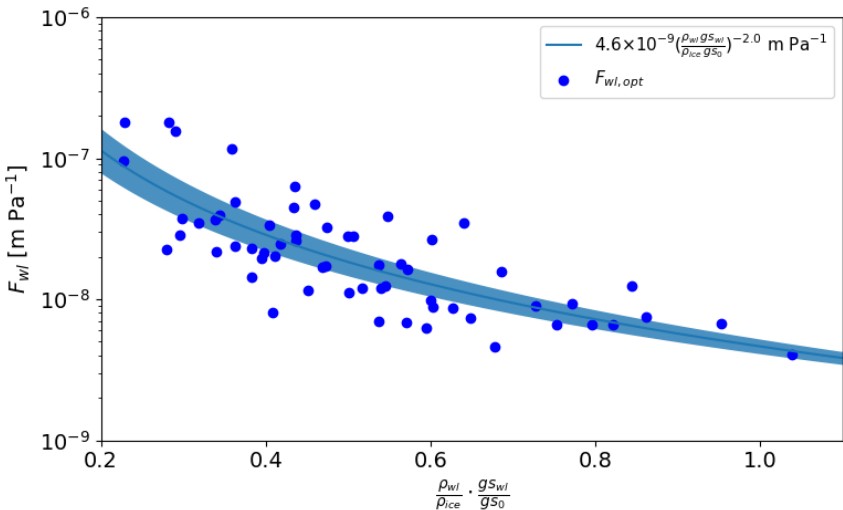

**Figure 7.** Parameter $F_{wl}^{opt}$ (Eq. (6)) with modeled normalized weak layer density $\frac{\rho_{wl}}{\rho_{ice}}$ times normalized grain size $\frac{gs_{wl}}{gs_0}$. The blue line shows a power law fit for $F_{wl}$ (Eq. (7)). Blue area is the 95% confidence interval of the 10-fold cross-validation.

the values of $F_{wl}$ to a power law function

$$F_{wl} = a \left( \left( \frac{\rho_{wl}}{\rho_{ice}} \right)^x \left( \frac{gs_{wl}}{gs_0} \right)^y \right)^b \tag{7}$$

with the fit parameters $a$ and $b$. To normalize grain size we select $gs_0 = 0.00125$ m according to Schweizer et al. (2008). With $x$ and $y$ integers ranging from -3 to 3, we evaluated 48 fit functions with regard to their ability of weak layer detection. Therefore,

we calculated DR (Eq. (4)) and MR (Eq. (5)) values for all field days ($m = 49$). The best performance, i.e. high DR and low MR, was obtained with $x = y = 1$, namely a DR of 0.89 and a MR of 0.64 (Figure 8). The original parameterization of Gaume et al. (2017) performed poorly in terms of weak layer detection with a relatively low DR of 0.18 and a high MR of 0.95. To exemplify these differences, on 28 January 2015, using the original parameterization (Eq. (2)), only one (d=1) of the two tested weak layers (nd=1) was within the five weakest layers (Figure 9c), resulting in a DR of 0.5 and a MR of 0.8 for that single day.

In contrast, using the fit function with $x = y = 1$, both weak layers were detected within the first four weakest layers, resulting in a DR of 1 and a MR of 0.5 (Figure 9d). We therefore suggest a new parameterization of $r_c$, where $\frac{D_{wl}}{G_{wl}}$ in Eq. (2) is replaced by $F_{wl}$:

$$r_c = \sqrt{E' D_{sl} F_{wl}} \sqrt{\frac{2\tau_p}{\sigma_n}}, \tag{8}$$

where $F_{wl}$ is given by:

$$F_{wl} = a \left( \frac{\rho_{wl}}{\rho_{ice}} \cdot \frac{gs_{wl}}{gs_0} \right)^b \text{[m Pa}^{-1}\text{]}, \tag{9}$$

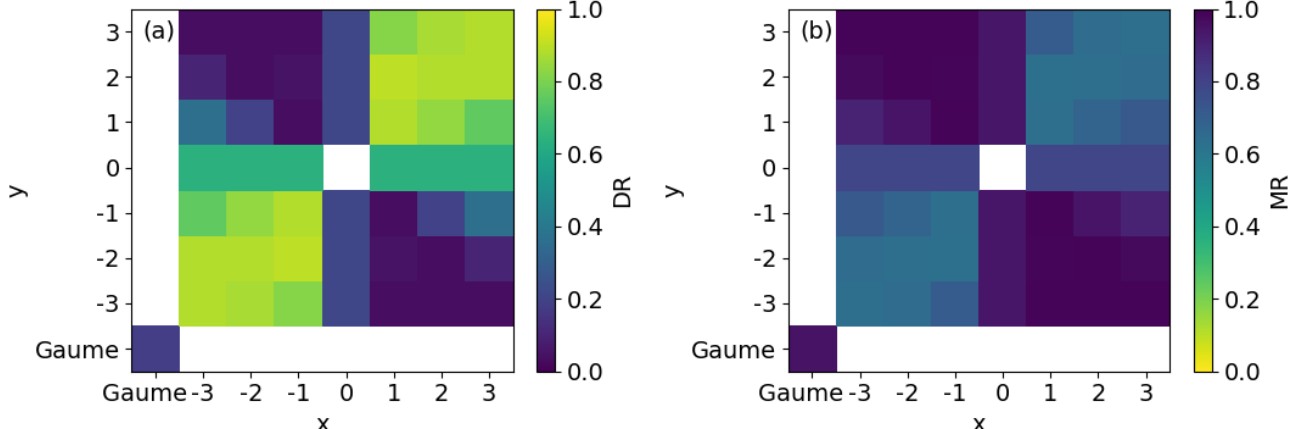

**Figure 8.** (a) Detection rate (DR) and (b) misclassification rate (MR) values for the 49 field days with exponents $x$ and $y$ for the power law fit function $F_{wl} = a\left(\rho_{wl}^x g s_{wl}^y\right)^b$. Values for x and y ranged from -3 to 3. The original parametrization of Gaume et al. (2017) (Eq. (2)) is shown in the lower left corner.

where $a = 4.6 \times 10^{-9} \pm 0.3 \times 10^{-9}$ m Pa$^{-1}$ and $b = -2.0 \pm 0.1$ are the mean fit parameters obtained with 10-fold cross-validation (Wilks, 2011). For this, we randomly split the joint data set into 10 groups, fitted $F_{wl}$ with nine groups and tested the fit function on the excluded group. After performing this ten times with each group serving as test group, we averaged the fit parameters and performance values. This yielded an average $NRMSE = 0.27 \pm 0.08$ for modeled $r_c$ from SNOWPACK
simulation using Eq. (8). Compared to the original parameterization (Eq. (2)) with an $NRMSE = 1.78$, the results highly improved. For SNOWPACK, values of $r_c$ ranged from 12 to 121 cm ($r_p = 0.90$, $p << 0.05$) using Eq. (8) (blue dots in Fig. 10). We also modeled $r_c$ values from manually observed snow profile data using the same fit factor (Equation (9)) in Eq. (8). The discrepancies between modeled values of critical crack length from manually observed snow profiles and measured $r_c$ values (orange dots in Fig. 10) were also removed using Eq. (8), with modeled $r_c$ values ranging from 2 to 156 cm ($r_p = 0.72$,
$p << 0.05$ and $NRMSE = 0.57$). Also, the match between observed and modeled time series of $r_c$ using SNOWPACK layer properties for the individual weak layers was substantially better when using Eq. (8) (Figure 11).

## 4  Discussion

We focused on validating the critical crack length parameterization in the snow cover model SNOWPACK. Crack propagation propensity only provides information on one of the processes required for avalanche release. Nevertheless, a critical weak
layer will likely have both a low failure initiation propensity and a low crack propagation propensity (Reuter and Schweizer, 2018). As such, we focused only on crack propagation, which is a fundamental process when assessing snow stability. The critical crack length provides valuable information on crack propagation (Gauthier and Jamieson, 2008b) and can directly be measured with PST experiments. Furthermore, PST experiments allow to directly compare measurements to the critical crack

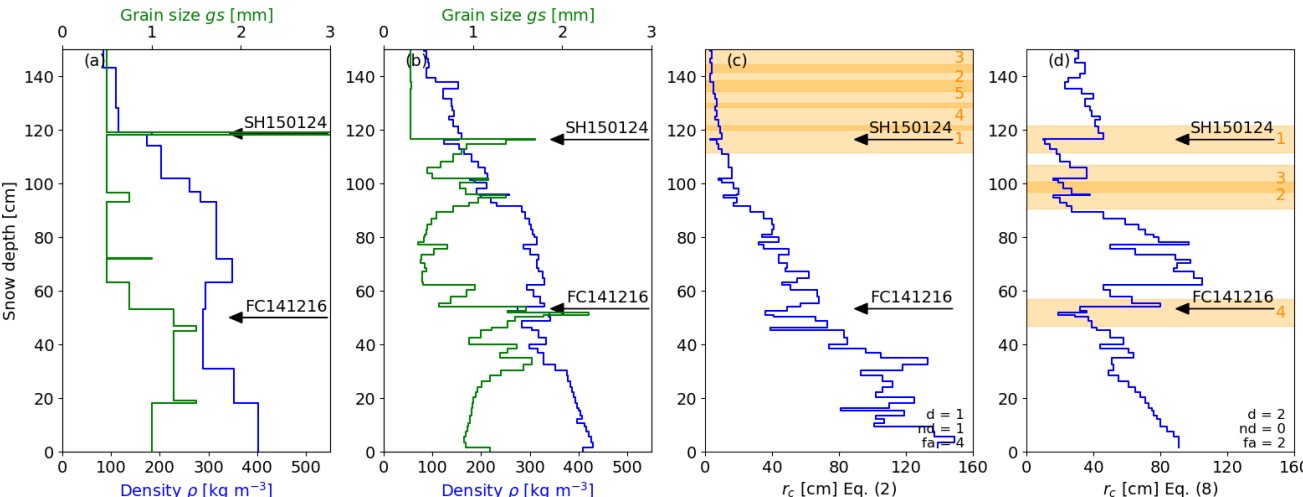

**Figure 9.** Observed (a) and simulated (b) density ($\rho$) and grain size ($gs$) profiles on 28 January 2015 at WAN7. Corresponding vertical $r_c$ profiles using layer properties from SNOWPACK for (c) the parameterization of Gaume et al. (2017) (Eq. (2)), and (d) the optimized parameterization (Eq. (8)). Arrows show the weak layers on which PST experiments were performed. Orange bars show the lowest values in the vertical $r_c$ profiles.

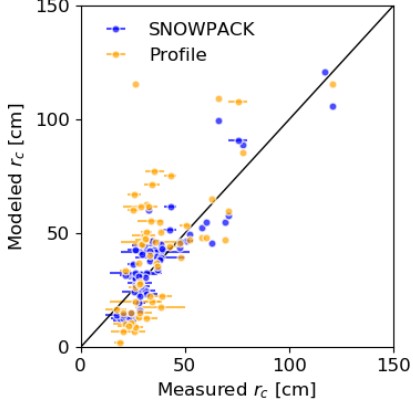

**Figure 10.** Modeled values of critical crack length (Eq. (8)) with layer properties from SNOWPACK (blue dots) and from manually observed profiles (orange dots) vs. averaged critical crack lengths from PST experiments. Error bars indicate the range of measured critical crack lengths for each point. The black line is the 1:1 line.

length modeled by SNOWPACK. This greatly facilitates the validation, especially when performing the measurements directly next to an automatic weather station used to drive SNOWPACK, as was done in this study. Due the vicinity to the AWS, no spatial interpolation was needed and the possible errors in the energy budget are assumed to be negligible. Indeed, for the field site WFJ we investigated the effect of different model configurations. For instance, using Dirichlet boundary conditions

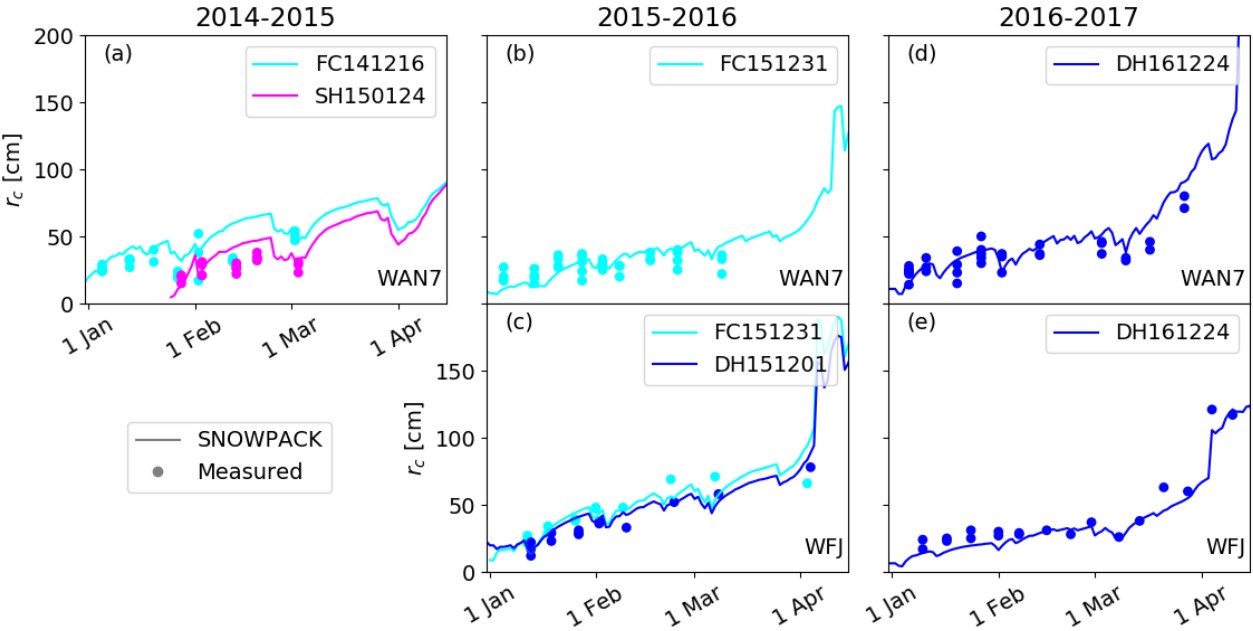

**Figure 11.** Evolution of the critical crack length at WAN7 (top) and WFJ (bottom) for the winter seasons 2014-2015 (a: WAN7), 2015-2016 (b: WAN7; c: WFJ) and 2016-2017 (d: WAN7; e: WFJ). Dots represent mean measured $r_c$ values from PST experiments, lines represent modeled $r_c$ values with layer properties from SNOWPACK using Eq. (8).

at the snow surface, i.e. directly using measured snow surface temperature, which did not influence our results, as differences in modeled snow properties were very small (not shown).

For the validation, we focused on prominent weak layers which were buried early in the season. Later in season, we also observed other failure layers in our CTs and ECTs. However, we did not perform PST experiments in these other failure layers,
as they did not show consistent crack propagation. Generally, PST experiments only provide a measure for the critical crack length in layers that are prone to crack propagation. Such layers are typically soft (hand hardness index $\leq 2$) and consist of rather large crystals, as typically found in the failure layers of avalanches (Schweizer and Jamieson, 2003; van Herwijnen and Jamieson, 2007). Performing a PST in other layers, such as for instance a layer of small rounded grains, generally does not yield any result for the critical crack length. The only micro-structural dependence in the original parameterization of Gaume et al.
(2017) (Eq. (2)) was through the grain type dependence of the shear strength $\tau_p$ developed by Jamieson and Johnston (2001) and implemented in SNOWPACK. Most of the observed weak layers consisted of faceted crystals and depth hoar crystals with measured densities of up to $366 \, \mathrm{kg \, m^{-3}}$. Differences in shear strength for rounded grains and faceted crystal as implemented in SNOWPACK are modest for densities around $300 \, \mathrm{kg \, m^{-3}}$. For densities above $330 \, \mathrm{kg \, m^{-3}}$ the shear strength of rounded grains even gets smaller than the shear strength of faceted crystal (Figure 2). This is a rather counter-intuitive, since rounded
grains are related to slab layers. Furthermore, most of the slab layers directly above weak layers also consist of faceted crystals

in both, observed and simulated profiles (Figure 3). Although weak layers and adjacent slab layers do not differ in grain type, they strongly differ in grain size (Figure 9a,b). It is clear, that SNOWPACK would ultimately benefit from micro-structural based parameterizations of shear strength and elastic modulus. However, currently only these rather simple formulations are available. Therefore, the modeled critical crack length as suggested by Gaume et al. (2017) mainly increased with depth
(Figure 9c), which was driven by the increase of density with depth. In contrast to density, grain size varied more prominently with weak layers often consisting of large grains (Figure 9a,b). Furthermore, modeled $r_c$ became unrealistically large late in the season (Figure 5). We therefore proposed a refined parameterization (Eq. (8), which strongly reduced the discrepancies between modeled and simulated critical crack lengths. Our refined parameterization greatly improved the results as it removed two variables of the original parameterization (Eq. (2)), which were not sufficiently well represented in SNOWPACK.

The first variable was weak layer thickness $D_{wl}$. The large discrepancies between observed and modeled $D_{wl}$ showed that simply using modeled $D_{wl}$ results in poor estimates of $r_c$ (Figure 4b). While a layer per definition should differ from adjacent layers in density or microstructure (Fierz et al., 2009), layer thicknesses are constrained by numerical stability in SNOWPACK. The simulation time step was 15 min and therefore snow layer thicknesses were restricted to approximately 3 cm. In contrast, observers define layer boundaries with some subjectivity and layer thicknesses of up to 30 cm were recorded in manually
observed snow profiles. To develop the original $r_c$ parameterization (Eq. (2)), Gaume et al. (2017) performed numerical simulations using an idealized structure of the weak layer and $D_{wl}$ was closely linked to collapse height. Indeed, when $r_c$ is reached in a PST experiment, crack propagation occurs inducing the structural collapse of the weak layer (e.g. van Herwijnen and Jamieson, 2005; van Herwijnen et al., 2010). The collapse height is believed to contribute to extensive fracture propagation (Jamieson and Schweizer, 2000; van Herwijnen and Jamieson, 2005; van Herwijnen et al., 2010). However, collapse heights
are generally around 1 to 10 mm in real PST experiments (van Herwijnen and Jamieson, 2005), i.e. on the order of the grain size rather than $D_{wl}$. While thus far it remains unclear whether the collapse height relates to $r_c$ and how it scales with grain size, it is plausible to consider grain size rather than weak layer thickness in the parameterization. Moreover, structural length, crystal size and grain size have been previously introduced to improve the paramterizations of mechanical properties (e.g. Proksch et al., 2015; Schulson, 2001; Schweizer et al., 2004).

The second variable in Eq. (2) was the shear modulus of the weak layer $G_{wl}$. Thus far, there are very few measurements of $G_{wl}$ (Föhn et al., 1998; Reiweger et al., 2010) and therefore $G_{wl}$ was kept constant in the original parameterization. Nevertheless, one would expect $G_{wl}$ to increase with increasing density, similar to $E$ (Scapozza, 2004; van Herwijnen et al., 2016). This would in part compensate the exaggerated seasonal increase in modeled $r_c$ (Figure 12). In the absence of a sound $G_{wl}$ parameterization, replacing $G_{wl}$ with a term depending on $\rho_{wl}$ to model $r_c$ therefore seems plausible.

Thus, we replaced the poorly constrained $\frac{D_{wl}}{G_{wl}}$ term with Eq. (9). The overall dependency of $r_c$ on layer density through shear strength therefore decreased, since the exponent for weak layer density in the shear strength is positive, while the exponent in the fit parameter is negative. Instead, we introduced the grain size, resulting in lower values in $r_c$ for larger grains (Figure 9). Hence, the temporal increase of $r_c$ was less pronounced with increasing density, resulting in more realistic seasonal trends (Figure 11). Furthermore, the grain size dependence greatly improved the performance of using modeled critical crack length
for weak layer detection (Figure 8).

The critical crack length can be calculated for every simulated snow layer (Figure 9d). However, this does not mean that in each layer a crack will actually propagate. Currently, it is not possible to distinguish simulated snow layers with high propagation propensity from others. Furthermore, SNOWPACK simulates considerably more layers than observed due to the mismatch of layer thicknesses between simulated and observed snow profiles. We chose a relatively simple approach without requiring any layer matching to automatically detect weak layers based on low values for the critical crack length. An associated weak layer was counted as detected, if it was located within $\pm 5$ cm of the minimum in the vertical profile of critical crack length. This approach avoids detecting many weak layers within a small range. While it is clear that the threshold value of 5 cm above and below the minimum is somewhat subjective, we are confident that it is not a gross misrepresentation when assessing the stability of the snowpack. The optimized parameterization (Eq. (8)) increased the detection rate from 0.18 to 0.89 compared to Eq. (2), while decreasing the misclassification rate from 0.95 to 0.64. Hence, the refined parameterization for the critical crack length can properly represent observed results from snow stability tests and observed weaknesses often agreed with minima in the vertical profile of simulated crack length. This approach does not solve the weak layer detection problem, as this is a complex task (Schweizer et al., 2006; Monti et al., 2014). Instead, this approach shows the overall improvements of the refined parameterization and suggests that weak layer detection seems feasible, taking into account the critical crack length.

The seasonal evolution of $r_c$ simulated with SNOWPACK using Eq. (8) for WFJ 2015-2016 showed a general increase of $r_c$ for each layer (Figure 12). With increasing snow depth due to precipitation, $r_c$ for each layer temporally decreased (e.g. at the beginning of February and March). The weak base in the lower 40 cm of the snowpack, which was tracked with the PST experiments, consistently showed lower $r_c$ values than those within the overlying slab in the simulation. The simulation also showed weaknesses, which formed later in the season e.g. a layer that had formed on 31 January at the snow surface at around 120 cm. Although these layers might have been weak layers, they were not contained in our data set of PST experiments and therefore counted as false alarms.

## 5   Conclusions

During three winter seasons we monitored the evolution of the critical crack length $r_c$ with PST experiments in persistent weak layers at two field sites above Davos, Switzerland. On 49 days, we collected data on 7 distinct persistent weak layers including 157 PST experiments. Comparing observed to modeled critical crack length showed that the recently suggested model by Gaume et al. (2017) generally overestimated the observed critical crack length, especially later in the season. The discrepancies are likely related to the weak layer thickness $D_{wl}$ and the shear modulus of the weak layer $G_{wl}$.

We therefore suggested an improved parameterization including weak layer grain size and weak layer density instead of $D_{wl}$ and $G_{wl}$ (Eq. (8)). The grain size term in the improved $r_c$ parameterization (Eq. (8)) allowed us to implicitly account for snow microstructure. This resulted in lower $r_c$ values for layers with larger grains, in line with our field experience. This also improved the detection rate by simply comparing low values in simulated critical crack lengths with associated weak layers. The critical crack length can either be modeled with simulated layer properties from the snow cover model SNOWPACK, or from data from manual snow pit observations. In both cases, Eq. (8) greatly improved the match between observed and

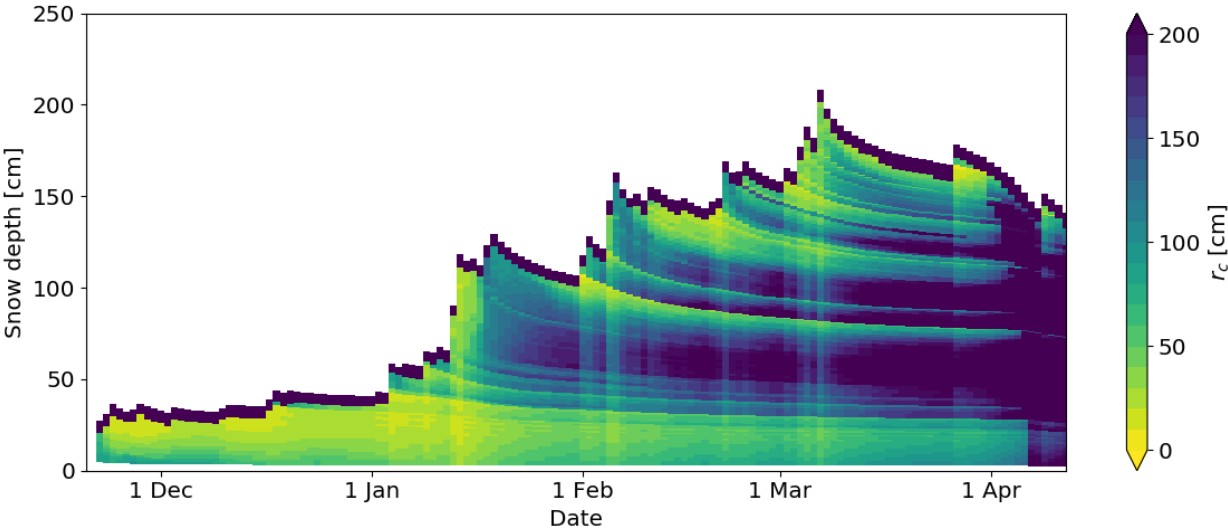

**Figure 12.** Temporal evolution of the vertical profile of the critical crack length modeled with SNOWPACK using Eq. (8) for WFJ 2015-2016.

modeled $r_c$ values and improved the representation of the observed seasonal evolution of the critical crack length. However, we want to highlight that the parameterization was developed based on data of weak layers of large faceted grains and could further be improved by sampling a greater diversity of weak layers.

The critical crack length relates to the onset of crack propagation and is therefore an important parameter to assess snow

5  stability. However, a snowpack is only prone to avalanche release if conditions for failure initiation and crack propagation are fulfilled. For the stability criteria in SNOWPACK, these conditions still need to be defined and verified with independent data. Clearly, the complex problem of automatically identifying weak layers and evaluating snow stability in simulated snow profiles is not yet solved. Nevertheless, our results are an encouraging step in the right direction.

*Data availability.* All relevant data, including manually observed snow profiles, stability tests and SNOWPACK simulations are available

10  on https://doi.org/10.16904/envidat.119.

*Code availability.* The numerical snow cover model SNOWPACK can be downloaded from http://models.slf.ch/p/snowpack/.

*Competing interests.* The authors declare that they have no conflict of interests.

*Acknowledgements.* We would like to thank all colleagues involved in the field campaigns, in particular Stephanie Mayer and Konstantin Nebel. We also like to thank two anonymous reviewers as well the Editor Guillaume Chambon, who helped us to improve this paper. Bettina Richter has been supported by a grant of the Swiss National Science Foundation (200021_169641).

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
