# Peer review of "Validating modeled critical crack length for crack propagation in the snow cover model SNOWPACK"

_The Cryosphere, 2019_

## Referee Comment (RC1) · Anonymous Referee #1 · 27 Jun 2019

General comments:

Crack propagation in a weak layer is a key process of slab avalanche release. The crack propagation propensity can be estimated via the critical crack length, i.e. the length of the saw cut in a PST at which the created crack will start propagating. This length can be directly measured in the field using PSTs but also computed from detailed snowpack profiles either measured in the field or simulated by snow cover models such as SNOWPACK.

This paper comprises one of the first direct evaluations of the critical crack length simulated by SNOWPACK with field measurements. The authors conducted their evaluation

on data spanning on two sites over two to three winter seasons in the region of Davos. On one hand, the authors use the parameterization proposed by Gaume et al. (2017) which relates the critical crack length to the weak layer shear elastic modulus, thickness and strength, and the slab weight, thickness and elastic modulus and applied it to SNOWPACK simulations forced by automatic weather stations. On the other hand, the authors used detailed snow stratigraphy and PST measurements performed on a weekly basis. The weak layer thickness in SNOWPACK is somehow related to numerical stability of the code (solver) and the history of the layer but has no direct relation to the weak layer collapse height, in contrast to the work of Gaume et al. (2017) used for the parameterization. The weak layer thickness measured in the field is related to the manual segmentation of the snowpack into layers and is generally larger than the "collapsible/active" part of the weak layer, due to the limited resolution of manual observations. Besides, the weak layer shear modulus has never been measured so far and must be assumed to be equal to 0.2 MPa. Therefore, the authors show that the direct application of the parameterization from Gaume et al. (2017) on the profiles simulated by SNOWPACK leads to poor agreement with the critical length measured in the field. As an alternative, the authors mainly propose to replace the "ambiguous" concept of weak layer thickness by a fitted function depending on weak layer grain size and density. With this new parameterization, the critical crack length is well reproduced by SNOWPACK with a normalized RMSE of 28%.

Overall, this study is very interesting as it tries to fill the gap between recent advances in snow mechanical modeling and numerical tools that can be used to forecast the avalanche danger. The results are new and exploit a huge, unique and valuable experimental dataset. Nevertheless, the fact that the initial parameterization of the critical crack length using the weak layer thickness as input leads to a poor agreement could have been presented before the final discussion. Indeed, for measured or simulated profiles, this thickness is more linked to the profile resolution rather than to the collapse height. In addition, adding a fitting factor would necessary improve the agreement between the simulations and the experiments. The methods and results are generally

clearly described. However, the presentation is sometimes clumsy and may benefit from consistent English editing (logic link between sentences, syntax) and some re-organization to make the reading more fluent. I recommend this study for publication pending presentation and content revisions (listed below).

Recommendation : Publication with Major Revisions

General comments/questions :

1) The authors first tried to apply the parameterization of Gaume et al. (2017) using the weak layer thickness of the SNOWPACK simulations or the thickness measured in the field. In Gaume et al. (2017) , the collapse height is directly linked to the weak layer thickness (assembly of spheres in a triangular shape). There is no reason why the resolution of the measured or simulated profile is related to the collapse height. This is effectively discussed in the paper (page 15-16) but too late in my opinion, which might mislead the reader (like me). Please consider some rewriting to announce this idea much earlier in the paper.

2) This article is mainly about crack propagation and compares the measured critical crack length to the simulated one. The experimental data comprises also CT and ECT. In the paper, it is not very clear to me how this specific data is used. I understand from line 4 page 7, that it is only used to detect the weak layer of interest but I feel that the data (Figure 3) is somehow unexploited or too detailed. Moreover, in section 2.2, the stability tests CT, ECT and PST are described with the same level of details. I suggest to reduce the description of the CT and ECT (or exploit it more) and give more details (scheme or photo, for instance) about the PST.

3) The authors associated the observed weak layer to a simulated weak layer based on their respective "birth" date. According to line 11-12 page 4, the "birth" date of simulated layers corresponds to the deposition date and the "birth" date of measured layers to their burial date. I don't understand why this should be the same. For instance, depth hoar might originate from shallow precipitation of the beginning of the

winter (date of birth) which progressively transformed into depth hoar under clear sky conditions and which was buried only some weeks after (date of burial). You need to clarify the matching method between the modeled and measured weak layers.

4) The weak layer density appears as a very important parameter of the critical crack length evaluation. Measuring density of thin and very fragile layers is challenging. Could you please add details on the measurement procedure (e.g. size of the cutter, etc.) and discuss some discrepancies (or no) that may originate from the limited vertical resolution of the cutter (compared to the thickness of the "active" weak layer part).

5) The model was evaluated in terms of probability of detection of the weak layer. The description of the methodology was not clear to me. First, I understood that the weak layer matching was already done with the "birth" date so you may only check whether the global minimum is located close to the associated simulated weak layer? Besides, as explained in the introduction, the stability of the weak layer-slab system is not only controlled by crack propagation propensity but also the sensitivity to trigger a crack (initiation). As the tracked weak layers were also identified by CTs, is it not hopeless to try to identify the observed weak layer only with the critical crack length? Last, it is not really clear to me how the probability of detection is computed. Does it mean that the weak layer is considered as detected when it is located in a band of 10 cm next to five "local" minima (i.e. an overall band of 50 to 30 cm)? Moreover, the term local minimum might be misleading as local minimum already refer to something well-defined (local minimum) and not the fact to "delete" a band of 10 cm in the search of iterative global minimum. According to Fig. 9d, you might consider to rank the real local minima by their prominence.

6) You use Neumann boundary conditions (heat flux imposed?). At WFJ, you also have the possibility to force the surface temperature, don't you? May this a way to get rid of possible errors in the surface energy budget that may cause discrepancies between the measured and simulated r_c, independently of the accuracy of the proposed parameterization? Indeed, you pointed some error (l. 4-8, p. 9) due to the presence/absence

of melt crusts. Add some discussion on this point.

Technical comments :

1) The abstract needs significant rewriting. It is too approximate and does not give a precise idea of the results. I listed some problems hereafter. The term "data" is used in the text but it is not clear to what it refers (measurements?). I do not get the logic of the sentence "especially if they also provide information on snow instability". The quantification of stability in terms of initiation, propagation, gliding is never presented. The reader may not understand that r_c is a measure propagation propensity. What was monitored in the experiments? What are the "two variables" (l. 6)? The word PST does not appear in the abstract, although it is the key measurement? The "NRMSE" (l. 8) of what ? What about the role of weak layer density? The algorithm of detection is not "simple". One sentence on the implications of this study is missing.

2) "snow instability tests" (l.3, p.2 and elsewhere). Please use everywhere where possible "stability" instead of "instability".

3) l.19, p. 1: Final gliding on the substrate may be added in the key processes of slab avalanche release.

4) l.20, p.1: "A third criterion". The first and second criteria were not defined in the text here. Besides, the slab propagation support should be presented as a second complementary criterion (in addition to r>r_c) for crack propagation.

5) l.21, p.1: "type and location" are not "questions".

6) l.3, p.2: "data" Do you mean measurements?

7) l.5, p.5: "can only be made". Too definitive. You can also do more experiments. Reword. "Numerical snow cover model can help increasing spatial and temporal resolution of ..."

8) l.8, p.2: "SCM predicts indices describing the avalanche danger at regional scale".

9) l.22, p.2: "good agreement". Can you be more precise?

10) l.27, p.2: "one type of weak layer". Which one?

11) l.31-33, p.2: the role of weak layer density is also re-inforced by the new parama-terization.

12) l.28, p.3: the mean $r\_c$ value of one to five PST tests is used. Why? It migth be worth to show the scatter (error bar?) on Figs. 6 and 10. Besides, individual $r\_c$ points are already shown on Figs. 5 and 11.

13) l.8, p.4: "was written for evey day". written -> stored. Can you give details on the exact time (eg. 6:00 UTC) of profile data? Can the comparison to measurements be affected by daily variation?

14) l.20-25, p.4: The shear strength of snow (except SH) is derived from power-law functions of density. Is it the standard of SNOWPACK, or is it a new parameterization? Give details/references.

15) Figure 2: Use international hand hardness code (Fierz et al., 2009; F, 1F, 4F, P, K, I) and explain the meaning in the legend. Is the shown total depth measured or simulated? I suggest to separate (a, b, c) from (d-i) into distinct figures and to SIGNIFICANTLY increase the vertical size of (d-i) and add the same graphs of the stratigraphy for WFJ. Moreover, could you add the density profile on the graphs.

16) Figure 4: I suggest to make a distinct large subplot for the graphs showing $D\_wl\_measured = f(D\_wl\_simulated)$

17) l.6, p.9: "observed weak layers [...] present in the simulated profiles". Currently it is not possible to see SH150124 in the measured profiles (no SH visible in Fig.2 e).

18) l.10, p.9 "Modeled slab". Could you detail somewhere how the slab is defined i.e. all layers above the weak layer (?).

19) l;6-8, p.9: "In the winter, ... degrees". I don't see a crust in Fig.2f ??? You described

one specific difference between the measured and simulated profile. There are other differences, why did you point this specific one out?

20) Figure 6: Enlarge the figures end use smaller dots for the points.

21) Figure 8: May it possible to express the results in terms of True Skill Score (TSS)?

22) Section 3.4. As far as I understand, the fitting is conducted on the SNOWPACK output and then also applied on the measured profile. Is that correct? Could you please clarify in the text.

23) l.7 p.13 to l.3 p.14: The first paragraph of the discussion belong to the introduction as it is not based on any result presented in this paper.

24) l.4 p16 "while thus far it remains unclear whether the collapse height relates to r_c". Could you give some references on this point? And add some expected trend from the literature?

---

## Referee Comment (RC2) · Anonymous Referee #2 · 31 Jul 2019

In this study, the authors seek to 1) investigate the performance and limitations of SNOWPACK for predicting the critical crack length using the previously suggested parameterization for the critical crack length from Gaume et al. 2017 and 2) potentially improve this performance with the introduction of a further-developed parameterization of the critical crack length.

With their improved parameterization, the authors demonstrate a significant enhancement of the performance of SNOWPACK in its ability to automatically detect the formation of weak layers in the snowpack. This is demonstrated through the thorough evaluation of the improved parameterization when compared to field-collected snowpit

data that was taken over 3 winter periods, at two locations, and with 145 propagation saw test experiments on known weak layers.

This work is thought to be of relevance to both the scientific research community as well as the operational avalanche forecasting community, and the authors are congratulated on their efforts.

After careful review, I have found that the article “Validating modeled critical crack length for crack propagation in the snow cover model SNOWPACK”, submitted to the The Cryosphere by Bettina Richter et al., is well-written, provides adequate evidence in support of their conclusions, and is in general suitable for publication with only minor revisions.

Recommendation: Publication with minor revisions

Detailed line item suggestions below:

Page 4, line 6: please briefly explain Neumann boundary conditions and why this was chosen for the snow surface.

Page 4, line 7: add citation for the chosen geothermal heat flux of 0.06 Wm-2.

Page 4, line 22: the density of the weak layer (rho_wl) does not yet appear to have been defined before being used inline in the text.

Page 5, figure 1: In this figure, please make clear in the text and caption where the a and b values came from or how they were derived.

Page 6, line 1-2: Can you comment or add a citation for how accurate these parameterizations are? Such that if it were possible to measure the weak layer shear strength and/or the elastic modulus of the slab in the field, should this be done? Or are these parameterizations thought to be adequate?

Page 6, line 16: why was a range of 5cm chosen?

Page 6, line 17: Curious, were there ever weak layers identified in the field that could not be tested with a PST test? (e.g. was the weak layer ever too thin or too difficult to follow with a saw blade?) Also, what are your general thoughts on the speed at which the saw blade is moved through the weak layer? Could this affect your results?

Page 9, line 13-15: perhaps you could further address this discrepancy in the weak layer thickness in the Discussion? Or briefly mention here that this was related to the boundary conditions chosen?

Page 13, Figure 8: I found the text to adequately describe the results and comparison to Gaume et al. 2017, would consider omitting this figure.

---

## Author Comment (AC1) · 9 Sep 2019

**Reply to Referee 1**

We thank the reviewer for the insightful and constructive comments. Below we will answer point by point.

**General comments/questions:**

1) The authors first tried to apply the parameterization of Gaume et al. (2017) using the weak layer thickness of the SNOWPACK simulations or the thickness measured in the field. In Gaume et al. (2017), the collapse height is directly linked to the weak layer

thickness (assembly of spheres in a triangular shape). There is no reason why the resolution of the measured or simulated profile is related to the collapse height. This is effectively discussed in the paper (page 15-16) but too late in my opinion, which might mislead the reader (like me). Please consider some rewriting to announce this idea much earlier in the paper.

It was certainly not our intention to mislead the reader in any way and we regret if this was the case. As suggested, we will introduce the close link between collapse height and weak layer thickness in the model of Gaume et al. (2017) earlier in the paper. We will mention the triangular shape of the weak layer in the work of Gaume et al. (2017) more explicitly in section 2.4 (Critical crack length parameterization). When describing differences in weak layer thickness between observation and SNOWPACK simulation in the last paragraph of section 3.2 (Modeled snow stratigraphy), we will explicitly link weak layer structure of Gaume et al. (2017) to collapse height.

2) This article is mainly about crack propagation and compares the measured critical crack length to the simulated one. The experimental data comprises also CT and ECT. In the paper, it is not very clear to me how this specific data is used. I understand from line 4 page 7, that it is only used to detect the weak layer of interest but I feel that the data (Figure 3) is somehow unexploited or too detailed. Moreover, in section 2.2, the stability tests CT, ECT and PST are described with the same level of details. I suggest to reduce the description of the CT and ECT (or exploit it more) and give more details (scheme or photo, for instance) about the PST.

As suggested, we will reduce the description of the snow instability tests (CT and ECT) in section 2.2 (Snow profiles and stability tests) and remove their results in Figure 3. Instead, we will add a photo with a schematic description of the PST in section 2.2.

3) The authors associated the observed weak layer to a simulated weak layer based on their respective "birth" date. According to line 11-12 page 4, the "birth" date of simulated layers corresponds to the deposition date and the "birth" date of measured
layers to their burial date. I don't understand why this should be the same. For instance, depth hoar might originate from shallow precipitation of the beginning of the winter (date of birth) which progressively transformed into depth hoar under clear sky conditions and which was buried only some weeks after (date of burial). You need to clarify the matching method between the modeled and measured weak layers.

We agree that the description for birth and deposition date of weak layers was not clear. We will clarify this issue in the revised manuscript. You are correct that for simulated snow layers the burial and deposition date are in general not the same. For observed weak layers we only know the burial date, i.e. the day when a weak snow surface was covered by new snow. Each modeled snow layer, however, was tagged within the SNOWPACK model with a deposition date corresponding to the date when a new layer was defined in the model. To match observed weak layers with the corresponding simulated layer, we therefore searched for the simulated layer, which was deposited immediately before the burial date of the observed weak layer. In other words, we identified the simulated weak layer by choosing the uppermost simulated layer with a deposition date older than the burial date of the observed weak layer. Layers of surface hoar are treated separately in SNOWPACK. Since surface hoar forms by deposition of water vapor from the air on the snow surface, and not from precipitation, it is only treated as snow layer within SNOWPACK, if certain conditions are fulfilled during burial. Thus, modeled surface hoar only "becomes" a snow layer at burial. Therefore, we first checked whether the observed layer of surface hoar (SH150124) was also modeled, i.e. buried within SNOWPACK on 24 January 2015. To temporally track this layer of modeled surface hoar, we identified the simulated weak layer by choosing the lowermost simulated layer with a deposition date equal to the burial date of the observed layer. We will accordingly clarify the description in the revised manuscript in section 2.3 (SNOWPACK).

4) The weak layer density appears as a very important parameter of the critical crack length evaluation. Measuring density of thin and very fragile layers is challenging.
Could you please add details on the measurement procedure (e.g. size of the cutter, etc.) and discuss some discrepancies (or no) that may originate from the limited vertical resolution of the cutter (compared to the thickness of the "active" weak layer part).

Manual measurements of density for layers thinner than about 3 cm are in fact not feasible. We will add more details about the type and size of the density cutters we used in section 2.2 (Snow profiles and stability tests) and discuss some discrepancies in section 4 (Discussion).

5) The model was evaluated in terms of probability of detection of the weak layer. The description of the methodology was not clear to me. First, I understood that the weak layer matching was already done with the "birth" date so you may only check whether the global minimum is located close to the associated simulated weak layer?

You are correct that we only investigated, how well weak layers can be detected based on the value of the critical crack length. Indeed, we checked whether the global minimum in the simulated vertical profiles of critical crack length was close to a simulated weak layer that was matched with the observations. Since we observed multiple weak layers in one snow profile, we decided to iteratively look for global minima by deleting a range of  $\pm$  5 cm around the prior global minima. We realize that the term "probability of detection" was misleading, since we do not present a method that allows doing so. Rather, a modeled crack length is assigned to every simulated layer in SNOWPACK. We then aim at finding weak layers based on low values of crack length. We will clarify the approach in the revised manuscript in section 2.5 (Model performance measures and weak layer detection) and change the term "probability of detection" to "detection rate" and change the term "false alarm ratio" to "misclassification rate".

Besides, as explained in the introduction, the stability of the weak layer-slab system is not only controlled by crack propagation propensity but also the sensitivity to trigger a crack (initiation). As the tracked weak layers were also identified by CTs, is it not hopeless to try to identify the observed weak layer only with the critical crack length?
We agree with the reviewer that crack propagation propensity only provides information on one of the processes required for avalanche release. Nevertheless, it is clear that a critical weak layer will have both a low failure initiation propensity and a low crack propagation propensity (Reuter and Schweizer, 2018). As such, focusing only on crack propagation still provides some information on stability. Furthermore, our goal was not to solve the weak layer detection problem, as various studies have shown that this is a very complex task (Schweizer et al., 2006, Monti, et al. 2014). We merely wanted to quantify the overall improvements of our modified critical crack length parameterization and highlight that it may be useful for weak layer detection. Our results suggest that this seems feasible. We will discuss this in more detail in section 4 (Discussion) of the revised manuscript.

Last, it is not really clear to me how the probability of detection is computed. Does it mean that the weak layer is considered as detected when it is located in a band of 10 cm next to five "local" minima (i.e. an overall band of 50 to 30 cm)? Moreover, the term local minimum might be misleading as local minimum already refer to something well-defined (local minimum) and not the fact to "delete" a band of 10 cm in the search of iterative global minimum.

Thanks for your careful review. You are correct that the term of local minima was misleading. In fact, we checked, whether an associated simulated weak layer was close to a global minimum of the critical crack length. Due to the mismatch of layer thicknesses between simulated and observed snow profiles, SNOWPACK produces considerably more layers than observed. We wanted to apply a relatively simple method, which would not require layer matching. You are correct that in our assessment, a weak layer was considered 'detected' if it was located within  $\pm$  5 cm of the minimum in the vertical profile of critical crack length. While it is clear that the threshold value of 5 cm above and below the minimum is somewhat subjective, we are confident that it is not a gross misrepresentation when assessing snow instability. As already mentioned above, we observed multiple weak layers within one snow profile and therefore decided
to check whether these associated weak layers were close to the five lowest values in the vertical profile of critical crack length. Looking for the five lowest values was also somewhat arbitrary but it allowed us to give a rate of false negative detections, which we will denote as "misclassification rate" in the revised manuscript. With this approach we believe that "deleting" a range of  $\pm$  5 cm is the only practicable method to avoid detecting within a small range too many weak layers that likely are not all relevant. We will clarify this in section 2.5 (Model performance measures and weak layer detection).

According to Fig. 9d, you might consider to rank the real local minima by their prominence.

We would rather not rank the real local minima by their prominence since this would require layer matching due to the mismatch in the number of layers in simulated profiles. We believe that our relatively simple approach is sufficient for the task at hand, and rather rank the 5 lowest critical crack lengths within the range of  $\pm$  5 cm in Fig. 9d.

6) You use Neumann boundary conditions (heat flux imposed?). At WFJ, you also have the possibility to force the surface temperature, don't you? May this a way to get rid of possible errors in the surface energy budget that may cause discrepancies between the measured and simulated  $r_c$ , independently of the accuracy of the proposed parameterization? Indeed, you pointed some error (l. 4-8, p. 9) due to the presence/absence of melt crusts. Add some discussion on this point.

You are right that we could have used snow surface temperature (TSS) at WFJ. At the field site WAN we could not use measured TSS as input to the SNOWPACK model, since the sensor was broken. To make the setup for the simulations consistent, we used Neumann boundary conditions to estimate TSS from energy fluxes (heat flux imposed). A comparison of Dirichlet boundary conditions (use measured TSS, Review Figure 1 bottom, at end) to Neumann boundary conditions (Review Figure 1 top, at end) at WFJ for winter season 2015-2016 did not show any considerable differences in simulated snow profiles.
We also compared observed with modeled critical crack lengths  $(r_c)$  using Neumann boundary conditions and Dirichlet boundary conditions (Review Figure 2, at end) for WFJ. Both simulations showed the same discrepancies between modeled and measured  $r_c$  for layer FC151231. For layer DH151201, the discrepancies were even higher using Dirichlet boundary conditions. On 20 January, modeled  $r_c$  for DH151201 (Dirichlet) jumped from 40 cm to 60 cm. In the simulation, this layer was merged with the one below, because differences in snow properties were low. Hence, the layer thickness increased from 0.9 cm to 1.8 cm due to the merging process. Since the original parameterization included layer thickness, this jump is an artifact of the merging of the layers. This even strengthens our assumption that modeled weak layer thickness is not a suitable variable to assess the critical crack length.

We will some discussion on the boundary conditions in the revised manuscript.

**Technical comments:**

1) The abstract needs significant rewriting. It is too approximate and does not give a precise idea of the results. I listed some problems hereafter. The term "data" is used in the text but it is not clear to what it refers (measurements?). I do not get the logic of the sentence "especially if they also provide information on snow instability". The quantification of stability in terms of initiation, propagation, gliding is never presented. The reader may not understand that r\_c is a measure propagation propensity. What was monitored in the experiments? What are the "two variables" (I. 6)? The word PST does not appear in the abstract, although it is the key measurement? The "NRMSE" (I.8) of what ? What about the role of weak layer density? The algorithm of detection is not "simple". One sentence on the implications of this study is missing.

We will revise the Abstract following these suggestions.

2) "snow instability tests" (I.3, p.2 and elsewhere). Please use everywhere where possible "stability" instead of "instability".

We will modify our wording and consistently avoid 'snow instability', and rather use

TCD
'snow stability tests' and 'stability indices'.

3) I.19, p. 1: Final gliding on the substrate may be added in the key processes of slab avalanche release.

You are correct to note that avalanche release ultimately depends on the friction between the slab and the substrate. We will more clearly describe the processes in dry-snow slab avalanche release.

4) I.20, p.1: "A third criterion". The first and second criteria were not defined in the text here. Besides, the slab propagation support should be presented as a second complementary criterion (in addition to  $r>r_c$ ) for crack propagation.

You are correct that we did not clearly define the first two criteria for avalanche release. We will more clearly describe the processes in dry-snow slab avalanche release.

5) I.21, p.1: "type and location" are not "questions".

You are correct to note that not type and location are questions, but that assessing snow instability requires information on the spatial distribution of weak layers found in a snowpack and what their triggering and propagation propensity are. To avoid any misunderstanding, we will more clearly describe the processes in dry-snow slab avalanche release.

6) I.3, p.2: "data" Do you mean measurements?

We will more clearly refer to the observations mentioned in the previous sentence.

7) I.5, p.5: "can only be made". Too definitive. You can also do more experiments. Reword. "Numerical snow cover model can help increasing spatial and temporal resolution of ..."

We are not aware of any efficient and feasible way to get a complete picture for regional forecasting by observations only. However, we will reword that sentence as suggested.

TCD
**We will change as suggested.**

9) I.22, p.2: "good agreement". Can you be more precise?

We will be more specific and mention that local minima in modeled critical crack lengths for one particular field day agreed with observed critical crack lengths."

10) I.27, p.2: "one type of weak layer". Which one?

We will mention that the weak layer consisted of faceted crystals.

11) I.31-33, p.2: the role of weak layer density is also reinforced by the new parameterization.

You are correct that the fit parameter contains both weak layer density and grain size. Furthermore, the shear strength of the weak layer also depends on weak layer density. However, since the exponent for weak layer density in the fit parameter is negative, and the exponent for the shear strength is positive, overall the weak layer density dependence reduces.

12) I.28, p.3: the mean r\_c value of one to five PST tests is used. Why? It migth be worth to show the scatter (error bar?) on Figs. 6 and 10. Besides, individual r\_c points are already shown on Figs. 5 and 11.

We chose to show individual results of PST experiments in Figures 5 and 11 to highlight the variability of field experiments. To avoid cluttering in Figures 6 and 10 we averaged the critical crack lengths measured with PST experiments of one distinct weak layer at a specific day and compared it to the corresponding simulated layer. We will clarify this point in the revised text. Additionally, as suggested, we will include error bars.

13) I.8, p.4: "was written for every day". written -> stored. Can you give details on the exact time (eg. 6:00 UTC) of profile data? Can the comparison to measurements be affected by daily variation?

TCD
The validation is not affected by daily variations, which are generally rather small. Manual measurements were performed between 10 and 14 UTC, and output from the model was stored daily at 11 UTC. We will add information on storage time in section 2.3 SNOWPACK.

14) I.20-25, p.4: The shear strength of snow (except SH) is derived from power-law functions of density. Is it the standard of SNOWPACK, or is it a new parameterization? Give details/references.

These are in fact the standard parameterizations of the shear strength in SNOWPACK. The shear strength for layers consisting of rounded grains, precipitation particles, fragmented particles, faceted crystals and depth hoar was implemented in SNOWPACK according to Jamieson and Johnston (2001). For layers of surface hoar the parameterization of shear strength was described in Lehning et al. (2004). We will give more details on modeled shear strength in section 2.3 (SNOWPACK).

15) Figure 2: Use international hand hardness code (Fierz et al., 2009; F, 1F, 4F,P, K, I) and explain the meaning in the legend. Is the shown total depth measured or simulated? I suggest to separate (a, b, c) from (d-i) into distinct figures and to SIGNIFI-CANTLY increase the vertical size of (d-i) and add the same graphs of the stratigraphy for WFJ. Moreover, could you add the density profile on the graphs.

Thanks for spotting this typo. We will change FF to 1F in Figure 2 and explain it in the legend.

Snow depths in Figure 2 a), b) and c) are measured. We will also mention this in the caption.

With this figure, we wanted to provide an overview of the three seasons that are covered in this study, so that the reader gets an idea of manual and simulated snow profiles with a particular focus on the location of the weak layer that were tracked. We therefore prefer to keep this as one large figure. However, we will increase the vertical size of figures d-I, as suggested.
Since we collected data at WAN7 during all three seasons, and not at WFJ, we decided to only show data from that site. While we could also include an additional figure with profile data from WFJ, given the proximity of both field sites and the similarities in snowpack structure, we do not think such an extra figure would add much to the paper. Finally, since we compare density measurements and simulations in Figure 4, we choose not to add density curves to this Figure, as it would be come too cluttered.

16) Figure 4: I suggest to make a distinct large subplot for the graphs showing D\_wl\_measured = f(D\_wl\_simulated)

We will do so as suggested.

17) I.6, p.9: "observed weak layers [...] present in the simulated profiles". Currently it is not possible to see SH150124 in the measured profiles (no SH visible in Fig.2 e).

This layer is very thin and difficult to see in Fig. 2e in the manuscript. However, by enlarging the vertical size in Figure 2, the SH150124 should become better visible.

18) I.10, p.9 "Modeled slab". Could you detail somewhere how the slab is defined i.e. all layers above the weak layer (?).

We defined the slab as all layers above the weak layer. We will clarify this in the revised manuscript in section 2.3 (SNOWPACK).

19) I;6-8, p.9: "In the winter, ... degrees". I don't see a crust in Fig.2f ??? You described one specific difference between the measured and simulated profile. There are other differences, why did you point this specific one out?

Crusts are layers consisting of melt-freeze polycrystals and are given in red in Figure 2. There is a crust in the manually observed profile at a height of about 40 cm (Figure 2f), while it is not present in the SNOWPACK simulation (Figure 2e). We agree that there are obviously a number of other discrepancies between observed and simulated profiles. The one we highlight is close to the weak layer we followed. We will clarify this point in the revised manuscript and make it clear that we simply provide one example.
20) Figure 6: Enlarge the figures end use smaller dots for the points.

We will change as suggested.

21) Figure 8: May it possible to express the results in terms of True Skill Score (TSS)?

As explained above we cannot provide the probability of detection in the strict sense. Therefore, we can also not calculate the True Skill Score. To express the results in terms of True Skill Score would require a somewhat different approach. As explained above we arbitrarily chose to look at the 5 lowest values of in the vertical rc profile and assess if the weak layers that were tested in the field were also included in those. If we want to look at true and false negative prediction, we also need to include all other layers in the vertical profiles, requiring us to match layers observed in the field with those simulated in SNOWPACK. We will hence change the terms "probability of detection" and "false alarm ratio" in the revised manuscript to "detection ratio" and "misclassification ratio".

22) Section 3.4. As far as I understand, the fitting is conducted on the SNOWPACK output and then also applied on the measured profile. Is that correct? Could you please clarify in the text.

You are correct, the fitting was done with SNOWPACK data and then applied to both SNOWPACK and manual snow profile data. We believe that the part with the fitting is clearly stated in the text, e.g. on page 11, lines 11 and 14. We will additionally clarify that this fitting parameter was then applied to manual snow profile data on page 13, lines 1 and 2.

23) I.7 p.13 to I.3 p.14: The first paragraph of the discussion belong to the introduction as it is not based on any result presented in this paper.

We will remove this paragraph and simply provide some motivation why we only looked at crack propagation.

24) I.4 p16 "while thus far it remains unclear whether the collapse height relates to r\_c".
Could you give some references on this point? And add some expected trend from the literature?

To the best of our knowledge, we are not aware of any study that relates collapse height to critical crack length.
**Timeline View - WFJ2\_RUN2016\_dirichlet.pro**

---

## Author Comment (AC2) · 9 Sep 2019

**Reply to Referee 2**

We thank the reviewer for the insightful and constructive comments. Below we will answer point by point.

**Technical comments:**

Page 4, line 6: please briefly explain Neumann boundary conditions and why this was chosen for the snow surface.

To estimate snow surface temperature (TSS), SNOWPACK either directly uses measured values for TSS (Dirichlet boundary conditions) or estimates TSS from energy fluxes (Neumann boundary conditions). At the field site WAN7, the TSS sensor malfunctioned and we could not use TSS as input for SNOWPACK. While for WFJ we could have used measured TSS, we wanted to make the simulations consistent. Differences in the surface energy budget resulting from these two boundary conditions do not affect our results (see answer to reviewer 1).

Page 4, line 7: add citation for the chosen geothermal heat flux of $0.06 \, \mathrm{W \, m^{-2}}$.

We will add the following citations: Pollack et al. (1993) and Davies and Davies (2010).

Page 4, line 22: the density of the weak layer ($\rho_{wl}$) does not yet appear to have been defined before being used inline in the text.

Thank you for noticing this error. We will define the weak layer density ($\rho_{wl}$) before.

Page 5, figure 1: In this figure, please make clear in the text and caption where the a and b values came from or how they were derived.

We used the shear strength as it is implemented in SNOWPACK by default. The values of a and b for different grain types are those given in Jamieson and Johnston (2001, Table 8). They provided fits for the different grain types based on their extensive sets of shear frame measurements. We will clarify where these values come from and refer explicitly to Table 8 of Jamieson and Johnston (2001).

Page 6, line 1-2: Can you comment or add a citation for how accurate these parameterizations are? Such that if it were possible to measure the weak layer shear strength and/or the elastic modulus of the slab in the field, should this be done? Or are these parameterizations thought to be adequate?

The parameterization of the elastic modulus was derived based on laboratory measurements performed by Scapozza (2004). As suggested in Gaume et al. (2017), slab density was related to Young's modulus by a power law fit of that data. Scapozza (2004) reported a correlation coefficient $r^2$ of 0.9 for the original parameterization. The

parameterization of shear strength is based on field measurements and was related to density in Jamieson and Johnston (2001). They reported correlation coefficients $r^2$ of 0.31 to 0.54, depending on grain type. In the revised manuscript in section 2.3 (SNOWPACK), we will clarify where these parameterizations originate from. As can be seen from the correlation coefficients the parameterizations are adequate, clearly more for the laboratory than for the field measurements. While it is clear that direct field measurements of these quantities would improve the predictions of the model, such measurements are rather difficult to perform and time consuming, in particular there exist no reliable measurement for the elastic modulus of snow. Better estimates of shear strength and elastic modulus in terms of density and especially microstructure would definitely be useful. Ultimately, SNOWPACK would greatly benefit from micro-structural based parameterizations of shear strength and elastic modulus. At present, these rather simple parameterizations are the best possible available. We will also add some discussion on these parameterizations in section 4 (Discussion).

Page 6, line 16: why was a range of 5 cm chosen?

We checked whether the weak layers were close to the five lowest values in the simulated vertical profile of critical crack length ($r_c$). Given the vertical resolution of simulated snow layers, SNOWPACK produces many more layers than observed. We wanted to apply a relatively simple method, which would not require layer matching. Due to the differences in layer thickness between modeled and observed snow profiles, some of the five lowest values in the simulation would likely be very close to each other. Therefore, a weak layer was considered as "detected" if it was located within $\pm$ 5 cm of a minimum in the vertical profile of critical crack length. While it is clear that the threshold value of 5 cm above and below the minimum $r_c$ value is somewhat subjective, we are confident that it is not a gross misrepresentation when assessing snow instability. We will give more detail on this approach in section 2.5 (Model performance measures and weak layer detection).

Page 6, line 17: Curious, were there ever weak layers identified in the field that could

not be tested with a PST test? (e.g. was the weak layer ever too thin or too difficult to follow with a saw blade?) Also, what are your general thoughts on the speed at which the saw blade is moved through the weak layer? Could this affect your results?

We often observed fractures in weak layers while performing a CT and ECT that we could not test in a PST. Typically, these are weak layers that are surrounded by layers of very soft snow (e.g. new snow or very freshly buried surface hoar). In those cases, it is very difficult to visually identify the weak layer and stay in the weak layer with the snow saw during a PST. While in some cases it is possible to get PST results in such layers, generally the results will be very inconsistent. We will add some more discussion on PST experiments on page 15, lines 1-2. The speed of the snow blade does not significantly influence the results of the critical crack length as shown by van Herwijnen et al. (2016).

Page 9, line 13-15: perhaps you could further address this discrepancy in the weak layer thickness in the Discussion? Or briefly mention here that this was related to the boundary conditions chosen?

You are correct and we will address this again in the Discussion section. We will explicitly mention that the boundary conditions, i.e. the simulation time step in SNOWPACK, limit layer thickness to approximately 3 cm.

Page 13, Figure 8: I found the text to adequately describe the results and comparison to Gaume et al. 2017, would consider omitting this figure.

For clarity, we prefer to keep this Figure. It explicitly shows that the best performance for Fwl is obtained with values of the exponents x = y = 1, i.e. the simple product of grain size and density.

---

## Author Response (AR1)

**Reply to Referee #1**

We thank the reviewer for this thorough analysis of our work and for the insightful and constructive feedback, which helped us to improve the paper. Below we will answer point by point. The reviewer initial comments are written in black, our answer in blue and the corrections in the paper are highlighted in blue. The line numbers used in the answers correspond to those of the revised manuscript.

**General comments/questions:**

1) The authors first tried to apply the parameterization of Gaume et al. (2017) using the weak layer thickness of the SNOWPACK simulations or the thickness measured in the field. In Gaume et al. (2017), the collapse height is directly linked to the weak layer thickness (assembly of spheres in a triangular shape). There is no reason why the resolution of the measured or simulated profile is related to the collapse height. This is effectively discussed in the paper (page 15-16) but too late in my opinion, which might mislead the reader (like me). Please consider some rewriting to announce this idea much earlier in the paper.

It was certainly not our intention to mislead the reader in any way and we regret if this was the case. As suggested, we introduced the close link between collapse height and weak layer thickness in the model of Gaume et al. (2017) earlier in the paper. We mentioned the triangular shape of the weak layer in the work of Gaume et al. (2017) more explicitly in section 2.4 (Critical crack length parameterization), ll. 3-4, p.7 in the revised manuscript. When describing differences in weak layer thickness between observation and SNOWPACK simulation in the last paragraph of section 3.2 (Modeled snow stratigraphy), l.16, p.10, we explicitly linked weak layer structure of Gaume et al. (2017) to collapse height.

2) This article is mainly about crack propagation and compares the measured critical crack length to the simulated one. The experimental data comprises also CT and ECT. In the paper, it is not very clear to me how this specific data is used. I understand from line 4 page 7, that it is only used to detect the weak layer of interest but I feel that the data (Figure 3) is somehow unexploited or too detailed. Moreover, in section 2.2, the stability tests CT, ECT and PST are described with the same level of details. I suggest to reduce the description of the CT and ECT (or exploit it more) and give more details (scheme or photo, for instance) about the PST.

As suggested, we reduced the description of the snow instability tests (CT and ECT) in section 2.2 (Snow profiles and stability tests), l.27, p.3 and removed their results. Instead, we added a photo with a schematic description of the PST in section 2.2 (Figure 1, p.4).

3) The authors associated the observed weak layer to a simulated weak layer based on their respective birth date. According to line 11-12 page 4, the birth date of simulated layers corresponds to the deposition date and the birth date of measured layers to their

[Figure]

**Figure 1:** Snow cover evolution simluted by SNOWPACK at the field site WFJ for winter season 2015-2016 using Dirichlet boundary conditions (above) and Neumann boundary conditions (below) at the snow surface.

burial date. I dont understand why this should be the same. For instance, depth hoar might originate from shallow precipitation of the beginning of the winter (date of birth) which progressively transformed into depth hoar under clear sky conditions and which was buried only some weeks after (date of burial). You need to clarify the matching method between the modeled and measured weak layers.

We agree that the description for birth and deposition date of weak layers was not clear. We clarified this issue in the revised manuscript. You are correct that for simulated snow layers the burial and deposition date are in general not the same. For observed weak layers we only know the burial date, i.e. the day when a weak snow surface was covered by new snow. Each modeled snow layer, however, was tagged within the SNOWPACK model with a deposition date corresponding to the date when a new layer was defined in the model. To match observed weak layers with the corresponding simulated layer, we therefore searched for the simulated layer, which was deposited immediately before the burial date of the observed weak layer. In other words, we identified the simulated weak layer by choosing the uppermost simulated layer with a deposition date older than the burial date of the observed weak layer. Layers of surface hoar are treated separately in SNOWPACK. Since surface hoar forms by deposition of water vapor from

[Figure]

**Figure 2:** Temporal evolution of measured r_c (dots) with modeled r_c of two weak layers at WFJ 2015-2016 using Dirichlet boundary condition (dashed line) and Neumann boundary condition (solid line).

the air on the snow surface, and not from precipitation, it is only treated as snow layer within SNOWPACK, if certain conditions are fulfilled during burial. Thus, modeled surface hoar only "becomes" a snow layer at burial. Therefore, we first checked whether the observed layer of surface hoar (SH150124) was also modeled, i.e. buried within SNOWPACK on 24 January 2015. To temporally track this layer of modeled surface hoar, we identified the simulated weak layer by choosing the lowermost simulated layer with a deposition date equal to the burial date of the observed layer. We accordingly clarified the description in the revised manuscript in section 2.3 (SNOWPACK), ll. 15-27, p.5.

4) The weak layer density appears as a very important parameter of the critical crack length evaluation. Measuring density of thin and very fragile layers is challenging. Could you please add details on the measurement procedure (e.g. size of the cutter, etc.) and discuss some discrepancies (or no) that may originate from the limited vertical resolution of the cutter (compared to the thickness of the active weak layer part).

Manual measurements of density for layers thinner than about 3 cm are in fact not feasible. We added more details about the type and size of the density cutters we used in section 2.2 (Snow profiles and stability tests) ll. 21-24, p.3.

5) The model was evaluated in terms of probability of detection of the weak layer. The description of the methodology was not clear to me. First, I understood that the weak layer matching was already done with the birth date so you may only check whether the global minimum is located close to the associated simulated weak layer?

You are correct that we only investigated, how well weak layers can be detected based on the value of the critical crack length. Indeed, we checked whether the global minimum in

the simulated vertical profiles of critical crack length was close to a simulated weak layer that was matched with the observations. Since we observed multiple weak layers in one snow profile, we decided to iteratively look for global minima by deleting a range of $\pm$ 5 cm around the prior global minima. We realized that the term probability of detection was misleading, since we did not present a method that allows doing so. Rather, a modeled crack length is assigned to every simulated layer in SNOWPACK. We aimed at finding weak layers based on low values of crack length. We clarified the approach in the revised manuscript in section 2.5 (Model performance measures and weak layer detection; l.21, p.7 - l.5, p.8 and changed the term probability of detection to detection rate and change the term false alarm ratio to misclassification rate.

Besides, as explained in the introduction, the stability of the weak layer-slab system is not only controlled by crack propagation propensity but also the sensitivity to trigger a crack (initiation). As the tracked weak layers were also identified by CTs, is it not hopeless to try to identify the observed weak layer only with the critical crack length?

We agree with the reviewer that crack propagation propensity only provides information on one of the processes required for avalanche release. Nevertheless, it is clear that a critical weak layer will have both a low failure initiation propensity and a low crack propagation propensity (Reuter and Schweizer, 2018). As such, focusing only on crack propagation still provides some information on stability. Furthermore, our goal was not to solve the weak layer detection problem, as various studies have shown that this is a very complex task (Schweizer et al., 2006, Monti, et al. 2014). We merely wanted to quantify the overall improvements of our modified critical crack length parameterization and highlight that it may be useful for weak layer detection. Our results suggest that this seems feasible. We discussed this in more detail in section 4 (Discussion) of the revised manuscript, ll.12-14, p.18.

Last, it is not really clear to me how the probability of detection is computed. Does it mean that the weak layer is considered as detected when it is located in a band of 10 cm next to five local minima (i.e. an overall band of 50 to 30 cm)? Moreover, the term local minimum might be misleading as local minimum already refer to something well-defined (local minimum) and not the fact to delete a band of 10 cm in the search of iterative global minimum.

Thanks for your careful review. You are correct that the term of local minima was misleading. In fact, we checked, whether an associated simulated weak layer was close to a global minimum of the critical crack length. Due to the mismatch of layer thicknesses between simulated and observed snow profiles, SNOWPACK produces considerably more layers than observed. We wanted to apply a relatively simple method, which would not require layer matching. You are correct that in our assessment, a weak layer was considered detected if it was located within $\pm$ 5 cm of the minimum in the vertical profile of critical crack length. While it is clear that the threshold value of 5 cm above and below the minimum is somewhat subjective, we are confident that it is not a gross

misrepresentation when assessing snow instability. As already mentioned above, we observed multiple weak layers within one snow profile and therefore decided to check whether these associated weak layers were close to the five lowest values in the vertical profile of critical crack length. Looking for the five lowest values was also somewhat arbitrary but it allowed us to give a rate of false negative detections, which we denoted as misclassification rate in the revised manuscript. With this approach we believe that deleting a range of $\pm$ 5 cm is the only practicable method to avoid detecting too many weak layers within a small range that likely are not all relevant. We clarified this in section 2.5 (Model performance measures and weak layer detection), ll.24-29, p.7 and discussed this method in more detail in section 4 (Discussion), ll.2-9, p.18.

According to Fig. 9d, you might consider to rank the real local minima by their prominence.

We would rather not rank the real local minima by their prominence since this would require layer matching due to the mismatch in the number of layers in simulated profiles. We believe that our relatively simple approach is sufficient for the task at hand, and rather ranked the 5 lowest critical crack lengths within the range of $\pm$ 5 cm in Fig. 9c,d.

6) You use Neumann boundary conditions (heat flux imposed?). At WFJ, you also have the possibility to force the surface temperature, dont you? May this a way to get rid of possible errors in the surface energy budget that may cause discrepancies between the measured and simulated $r_c$, independently of the accuracy of the proposed parameterization? Indeed, you pointed some error (l. 4-8, p. 9) due to the presence/absence of melt crusts. Add some discussion on this point.

You are right that we could have used snow surface temperature (TSS) at WFJ. At the field site WAN we could not use measured TSS as input to the SNOWPACK model, since the sensor was broken. To make the setup for the simulations consistent, we used Neumann boundary conditions to estimate TSS from energy fluxes (heat flux imposed). A comparison of Dirichlet boundary conditions (use measured TSS, Review Figure 1 bottom, at end) to Neumann boundary conditions (Review Figure 1 top, at end) at WFJ for winter season 2015-2016 did not show any considerable differences in simulated snow profiles.
We also compared observed with modeled critical crack lengths ($r_c$) using Neumann boundary conditions and Dirichlet boundary conditions (Review Figure 2, at end) for WFJ. Both simulations showed the same discrepancies between modeled and measured $r_c$ for layer FC151231. For layer DH151201, the discrepancies were even higher using Dirichlet boundary conditions. On 20 January, modeled $r_c$ for DH151201 (Dirichlet) jumped from 40 cm to 60 cm. In the simulation, this layer was merged with the one below, because differences in snow properties were low. Hence, the layer thickness increased from 0.9 cm to 1.8 cm due to the merging process. Since the original parameterization included layer thickness, this jump is an artifact of the merging of the layers. This even strengthens our assumption that modeled weak layer thickness is not a suitable variable

to assess the critical crack length.
We added some discussion on the boundary conditions in the revised manuscript in section 4 (Discussion), ll.1-2, p.16.

**Technical comments:**

1) The abstract needs significant rewriting. It is too approximate and does not give a precise idea of the results. I listed some problems hereafter. The term data is used in the text but it is not clear to what it refers (measurements?). I do not get the logic of the sentence especially if they also provide information on snow instability. The quantification of stability in terms of initiation, propagation, gliding is never presented. The reader may not understand that r_c is a measure propagation propensity. What was monitored in the experiments? What are the two variables (l. 6)? The word PST does not appear in the abstract, although it is the key measurement? The NRMSE (l.8) of what ? What about the role of weak layer density? The algorithm of detection is not simple. One sentence on the implications of this study is missing.

We revised the Abstract following these suggestions.

2) snow instability tests (l.3, p.2 and elsewhere). Please use everywhere where possible stability instead of instability.

We modified our wording and consistently avoid snow instability, and rather used snow stability tests and stability indices.

3) l.19, p.1: Final gliding on the substrate may be added in the key processes of slab avalanche release.

You are correct to note that avalanche release ultimately depends on the friction between the slab and the substrate. We described the processes of dry-snow slab avalanche release more clearly in section 1 (Introduction), ll.1-2, p.2.

4) l.20, p.1: A third criterion. The first and second criteria were not defined in the text here. Besides, the slab propagation support should be presented as a second complementary criterion (in addition to r¿r_c) for crack propagation.

You are correct that we did not clearly define the first two criteria for avalanche release. We described the processes in dry-snow slab avalanche release more clearly in section 1 (Introduction) in l.22, p.1 - l.2, p.2.

5) l.21, p.1: type and location are not questions.

You are correct to note that not type and location are questions, but that assessing snow instability requires information on the spatial distribution of weak layers found in a snowpack and what their triggering and propagation propensity are. We clarified this in ll.3-4, p.2.

6) l.3, p.2: data Do you mean measurements?

To avoid any misunderstandings, we refered to the observations mentioned in the previous sentence (l.7, p.2).

7) l.5, p.5: can only be made. Too definitive. You can also do more experiments. Reword. Numerical snow cover model can help increasing spatial and temporal resolution of ...

We are not aware of any efficient and feasible way to get a complete picture for regional forecasting by observations only. However, we reworded that sentence as suggested.

8) l.8, p.2: SCM predicts indices describing the avalanche danger at regional scale.

We changed l.12, p.2 in the revised manuscript as suggested.

9) l.22, p.2: good agreement. Can you be more precise?

We described in more detail that local minima in modeled critical crack lengths for one particular field day agreed with observed critical crack lengths in l.34, p.2.

10) l.27, p.2: one type of weak layer. Which one?

We mentioned that the weak layer consisted of faceted crystals in l.4, p.3.

11) l.31-33, p.2: the role of weak layer density is also reinforced by the new parameterization.

You are correct that the fit parameter contains both weak layer density and grain size. Furthermore, the shear strength of the weak layer also depends on weak layer density. However, since the exponent for weak layer density in the fit parameter is negative, and the exponent for the shear strength is positive, overall the weak layer density dependence reduces. We mentioned the reduced dependency on weak layer density in section 1, l.10, p.3 and section 4, ll.30-32, p.17.

12) l.28, p.3: the mean r_c value of one to five PST tests is used. Why? It migth be worth to show the scatter (error bar?) on Figs. 6 and 10. Besides, individual r_c points are already shown on Figs. 5 and 11.

We chose to show individual results of PST experiments in Figures 5 and 11 to highlight the variability of field experiments. To avoid cluttering in Figures 6 and 10 we averaged the critical crack lengths measured with PST experiments of one distinct weak layer at a specific day and compared it to the corresponding simulated layer. We clarified this point in the revised text in section 3.3 (Evolution of the critical crack length), ll.3-4, p.11. Additionally, we included the range of of measurements in Figures 6 and 10.

13) l.8, p.4: "was written for every day". written -¿ stored. Can you give details on the

exact time (eg. 6:00 UTC) of profile data? Can the comparison to measurements be affected by daily variation?

The validation is not affected by daily variations, which are generally rather small. Manual measurements were performed between 10 and 14 UTC, and output from the model was stored daily at 11 UTC. We added information on storage time in section 2.3 (SNOWPACK), ll.10-12, p.5.

14) l.20-25, p.4: The shear strength of snow (except SH) is derived from power-law functions of density. Is it the standard of SNOWPACK, or is it a new parameterization? Give details/references.

These are in fact the standard parameterizations of the shear strength in SNOWPACK. The shear strength for layers consisting of rounded grains, precipitation particles, fragmented particles, faceted crystals and depth hoar was implemented in SNOWPACK according to Jamieson and Johnston (2001). For layers of surface hoar the parameterization of shear strength was described in Lehning et al. (2004). We gave more details on modeled shear strength in section 2.3 (SNOWPACK), ll.2-6, p.6.

15) Figure 2: Use international hand hardness code (Fierz et al., 2009; F, 1F, 4F,P, K, I) and explain the meaning in the legend. Is the shown total depth measured or simulated? I suggest to separate (a, b, c) from (d-i) into distinct figures and to SIGNIFICANTLY increase the vertical size of (d-i) and add the same graphs of the stratigraphy for WFJ. Moreover, could you add the density profile on the graphs.

Thanks for spotting this typo. We changed FF to 1F in Figure 3 and explained the hand hardness code in the legend.
Snow depths in Figure 3 a), b) and c) are measured. We also mentioned this in the caption.
With this figure, we wanted to provide an overview of the three seasons that are covered in this study, so that the reader gets an idea of manual and simulated snow profiles with a particular focus on the location of the weak layer that were tracked. We therefore kept this as one large figure. However, we increased the vertical size of figures d-i, as suggested.
Since we collected data at WAN7 during all three seasons, and not at WFJ, we decided to only show data from that site. While we could also include an additional figure with profile data from WFJ, given the proximity of both field sites and the similarities in snowpack structure, we do not think such an extra figure would add much to the paper. Finally, since we compared density measurements and simulations in Figure 4, we chose not to add density curves to this Figure, as it would be come too cluttered. Instead, we added density and grain size profiles on 28 January 2015 to Figure 9 and dicussed these profiles in more detail in section 4, ll.2-6, p.17.

16) Figure 4: I suggest to make a distinct large subplot for the graphs showing D_wl_measured = f(D_wl_simulated)

We changed Figure 4 as suggested.

17) l.6, p.9: "observed weak layers [...] present in the simulated profiles". Currently it is not possible to see SH150124 in the measured profiles (no SH visible in Fig.2 e).

We enlarged the vertical size in Figure 3 and the SH150124 became better visible in Figure 3d,e.

18) l.10, p.9 "Modeled slab". Could you detail somewhere how the slab is defined i.e. all layers above the weak layer (?).

We defined the slab as all layers above the weak layer. We clarified this in the revised manuscript in section 2.3 (SNOWPACK), ll.27-29, p.5.

19) l;6-8, p.9: "In the winter, ... degrees". I dont see a crust in Fig.2f ??? You described one specific difference between the measured and simulated profile. There are other differences, why did you point this specific one out?

Crusts are layers consisting of melt-freeze polycrystals and are given in red in Figure 3. There is a crust in the manually observed profile at a height of about 40 cm (Figure 3f), while it is not present in the SNOWPACK simulation (Figure 3g). We agree that there are obviously a number of other discrepancies between observed and simulated profiles. The one we highlighted is close to the weak layer we followed. We clarified this point in the revised manuscript in section 3.2 (Modeled snow stratigraphy), ll.4-6, p.10 and made it clear that we simply provided one example.

20) Figure 6: Enlarge the figures end use smaller dots for the points.

We changed as suggested.

21) Figure 8: May it possible to express the results in terms of True Skill Score (TSS)?

As explained above we cannot provide the probability of detection in the strict sense. Therefore, we can also not calculate the True Skill Score. To express the results in terms of True Skill Score would require a somewhat different approach. As explained above we arbitrarily chose to look at the 5 lowest values of in the vertical rc profile and assess if the weak layers that were tested in the field were also included in those. If we want to look at true and false negative prediction, we also need to include all other layers in the vertical profiles, requiring us to match layers observed in the field with those simulated in SNOWPACK. We hence changed the terms probability of detection and false alarm ratio in the revised manuscript to detection ratio and misclassification ratio.

22) Section 3.4. As far as I understand, the fitting is conducted on the SNOWPACK output and then also applied on the measured profile. Is that correct? Could you please clarify in the text.

You are correct, the fitting was done with SNOWPACK data and then applied to both SNOWPACK and manual snow profile data. We believe that the part with the fitting is clearly stated in the text, e.g. on l.30 and l. 33, p.11. We additionally clarified that this fitting parameter was then applied to manual snow profile data in section 3.4 (Improvements to $r_c$ parameterization l.7, p.14.

23) l.7 p.13 to l.3 p.14: The first paragraph of the discussion belong to the introduction as it is not based on any result presented in this paper.

We removed this paragraph and simply provided some motivation why we only looked at crack propagation in section 4, ll.13-16, p.14.

24) l.4 p16 "while thus far it remains unclear whether the collapse height relates to r_c". Could you give some references on this point? And add some expected trend from the literature?

To the best of our knowledge, we are not aware of any study that relates collapse height to critical crack length.

**Reply to Referee #2**

We would like to thank the reviewer for the insightful and the positive and constructive feedback, which helped us to improve the paper. Below we will answer point by point. The reviewer initial comments are written in black, our answer in blue and the corrections in the paper are highlighted in blue. The line numbers used in the answers correspond to those of the revised manuscript.

**Technical comments:**

Page 4, line 6: please briefly explain Neumann boundary conditions and why this was chosen for the snow surface.

To estimate snow surface temperature (TSS), SNOWPACK either directly uses measured values for TSS (Dirichlet boundary conditions) or estimates TSS from energy fluxes (Neumann boundary conditions). At the field site WAN7, the TSS sensor malfunctioned and we could not use TSS as input for SNOWPACK. While for WFJ we could have used measured TSS, we wanted to make the simulations consistent. Differences in the surface energy budget resulting from these two boundary conditions did not affect our results (see answer to reviewer #1). We added some discussion in section 4, ll.1-2, p.16.

Page 4, line 7: add citation for the chosen geothermal heat flux of 0.06 $\mathrm{W\,m^{-2}}$.

We added the following citations: Pollack et al. (1993) and Davies and Davies (2010) in section 2.3 (SNOWPACK), l.9, p.5.

Page 4, line 22: the density of the weak layer ($\rho_{wl}$) does not yet appear to have been defined before being used inline in the text.

Thank you for noticing this error. We defined the weak layer density ($\rho_{wl}$) before in section 2.3 , l.4, p.6 in the revised manuscript.

Page 5, figure 1: In this figure, please make clear in the text and caption where the a and b values came from or how they were derived.

We used the shear strength as it is implemented in SNOWPACK by default. The values of a and b for different grain types are those given in Jamieson and Johnston (2001, Table 8). They provided fits for the different grain types based on their extensive sets of shear frame measurements. We clarified where these values come from and refered explicitly to Table 8 of Jamieson and Johnston (2001) in l.6, p.6.

Page 6, line 1-2: Can you comment or add a citation for how accurate these parameterizations are? Such that if it were possible to measure the weak layer shear strength and/or the elastic modulus of the slab in the field, should this be done? Or are these parameterizations thought to be adequate?

The parameterization of the elastic modulus was derived based on laboratory measurements performed by Scapozza (2004). As suggested in Gaume et al. (2017), slab density was related to Youngs modulus by a power law fit of that data. Scapozza (2004) reported a correlation coefficient $r^2$ of 0.9 for the original parameterization. The parameterization of shear strength is based on field measurements and was related to density in Jamieson and Johnston (2001). They reported correlation coefficients $r^2$ of 0.31 to 0.54, depending on grain type. In the revised manuscript in section 2.3 (SNOWPACK), we clarified where these parameterizations originate from (l.1 and l.5, p.6). As can be seen from the correlation coefficients the parameterizations are adequate, clearly more for the laboratory than for the field measurements. While it is clear that direct field measurements of these quantities would improve the predictions of the model, such measurements are rather difficult to perform and time consuming, in particular there exist no reliable measurement for the elastic modulus of snow. Better estimates of shear strength and elastic modulus in terms of density and especially microstructure would definitely be useful. Ultimately, SNOWPACK would greatly benefit from micro-structural based parameterizations of shear strength and elastic modulus. At present, these rather simple parameterizations are the best possible available. We also added some discussion on these parameterizations in section 4 (Discussion), l.12, p.16 - l.4, p.17.

Page 6, line 16: why was a range of 5 cm chosen?

We checked whether the weak layers were close to the five lowest values in the simulated vertical profile of critical crack length. Given the vertical resolution of simulated snow layers, SNOWPACK produces many more layers than observed. We wanted to apply a relatively simple method, which would not require layer matching. Due to the differences in layer thickness between modeled and observed snow profiles, some of the five lowest values in the simulation would likely be very close to each other. Therefore, a weak layer was considered as detected if it was located within $\pm$ 5 cm of a minimum in the vertical profile of critical crack length. While it is clear that the threshold value of 5 cm above and below the minimum rc value is somewhat subjective, we are confident that it is not a gross misrepresentation when assessing snow instability. We clarified this method in section 2.5, ll.22-29 and discussed this method in section 4, ll.1-9, p.18.

Page 6, line 17: Curious, were there ever weak layers identified in the field that could not be tested with a PST test? (e.g. was the weak layer ever too thin or too difficult to follow with a saw blade?) Also, what are your general thoughts on the speed at which the saw blade is moved through the weak layer? Could this affect your results?

We often observed fractures in weak layers while performing a CT and ECT that we could not test in a PST. Typically, these are weak layers that are surrounded by layers of very soft snow (e.g. new snow or very freshly buried surface hoar). In those cases, it is very difficult to visually identify the weak layer and stay in the weak layer with the snow saw during a PST. While in some cases it is possible to get PST results in such layers, generally the results will be very inconsistent. We added some more discussion

on PST experiments on ll.3-4, p.16. The speed of the snow blade does not significantly influence the results of the critical crack length as shown by van Herwijnen et al. (2016).

Page 9, line 13-15: perhaps you could further address this discrepancy in the weak layer thickness in the Discussion? Or briefly mention here that this was related to the boundary conditions chosen?

We addressed the discrepancy in thickness again in the Discussion section. We explicitly mentioned that the boundary conditions, i.e. the simulation time step in SNOWPACK, limit layer thickness to approximately 3 cm in ll.12-16, p.17.

Page 13, Figure 8: I found the text to adequately describe the results and comparison to Gaume et al. 2017, would consider omitting this figure.

For clarity, we prefered to keep this Figure. It explicitly showed that the best performance for Fwl is obtained with values of the exponents x = y = 1, i.e. the simple product of grain size and density.

[revised manuscript text omitted]